# An Axiomatic Theory of Provably-Fair Welfare-Centric Machine Learning

**Cyrus Cousins**
Department of Computer Science
Brown University
cyrus_cousins@brown.edu

## Abstract

We address an inherent difficulty in welfare-theoretic fair machine learning (ML), by proposing an equivalently-axiomatically justified alternative setting, and studying the resulting computational and statistical learning questions. Welfare metrics quantify *overall wellbeing* across a *population* of *groups*, and welfare-based objectives and constraints have recently been proposed to incentivize *fair ML methods* to satisfy their diverse needs. However, many ML problems are cast as *loss minimization* tasks, rather than *utility maximization*, and thus require nontrivial modeling to construct *utility functions*. We define a complementary metric, termed *malfare*, measuring overall *societal harm*, with axiomatic justification via the standard axioms of cardinal welfare, and cast fair ML as *malfare minimization* over the *risk values* (expected losses) of each group. Surprisingly, the axioms of cardinal welfare (malfare) dictate that this is not equivalent to simply defining utility as negative loss and maximizing welfare. Building upon these concepts, we define *fair-PAC learning*, where a fair-PAC learner is an algorithm that learns an $\varepsilon$-$\delta$ malfare-optimal model with bounded sample complexity, for *any data distribution* and (axiomatically justified) malfare concept. Finally, we show conditions under which many standard PAC-learners may be converted to fair-PAC learners, which places fair-PAC learning on firm theoretical ground, as it yields statistical — and in some cases computational — efficiency guarantees for many well-studied ML models. Fair-PAC learning is also practically relevant, as it democratizes fair ML by providing concrete training algorithms with rigorous generalization guarantees.

## 1 Introduction

It is now well-understood that contemporary ML systems exhibit differential accuracy across gender, race, and other protected groups, for tasks like facial recognition [5–7], in medical settings [2, 17], and many others. This exacerbates existing inequality, as facial recognition in policing yields disproportionate false-arrest rates, and medical ML yields disproportionate health outcomes. Welfare-centric ML methods encode both *accuracy* and *fairness* in a *welfare function* defined on a *set of groups*, and optimize or constrain welfare to *learn fairly*. This addresses differential accuracy and bias issues across groups by ensuring that (1) each group is *seen* and *considered* during training, and (2) an outcome is incentivized that is desirable overall, ideally according to some mutually-agreed-upon welfare function. Unfortunately, welfare metrics require a notion of (positive) utility, and we argue that this is not natural to many ML tasks, wherein we instead *minimize* some *negatively connoted* loss value. We thus define a complementary measure to *welfare*, which we term *malfare*, measuring societal harm (rather than wellbeing). In particular, malfare arises naturally when one applies the standard *axioms of cardinal welfare* to *loss values* rather than *utility values*. We then cast fair ML as a direct *malfare minimization* problem, and study its computational and statistical aspects.

35th Conference on Neural Information Processing Systems (NeurIPS 2021).

Perhaps surprisingly, minimizing a *malfare function* is *not equivalent* to maximizing some *welfare function* while taking utility to be negative loss, except in the egalitarian and utilitarian edge cases. This is because nearly every function satisfying the standard axioms of cardinal welfare requires *nonnegative* inputs, and it is in general impossible to contort a loss function into a utility function while satisfying this requirement. For example, while minimizing the 0-1 loss, which simply counts the number of mistakes a classifier makes, is isomorphic to maximizing the 1-0 gain, which counts number of correct classifications, minimizing some *malfare function* defined on 0-1 loss over groups *is not* in general equivalent to maximizing any *welfare function* defined on 1-0 gain.

Building upon these concepts, we develop a generic notion of fair ML, termed *fair-PAC* (FPAC) *learning*, where the goal is to learn models for which finite training samples may guarantee (additively) $\varepsilon$-optimal malfare, with probability at least $1 - \delta$, for *any* (axiomatically justified) malfare concept. This definition extends Valiant's [31] classic PAC-learning formalization, and we show that, with appropriate modifications, many (standard) PAC-learners may be converted to FPAC learners. In particular, we show via a constructive polynomial reduction that *realizable FPAC-learning* reduces to *realizable PAC-learning*. Furthermore, we show, non-constructively, that for learning problems where PAC-learnability implies uniform convergence, it is equivalent to FPAC-learnability. We also show that when training is possible via *convex optimization* under standard assumptions, then training $\varepsilon$-$\delta$ malfare-optimal models, like training risk-optimal models, requires polynomial time. We briefly summarize our contributions below.

1. We derive in section 2 the *malfare concept*, extending welfare to measure *negatively-connoted* attributes, and show that *malfare-minimization* naturally generalizes *risk-minimization* to produce *fairness-sensitive* ML objectives that consider multiple protected groups.
2. Section 3 extends PAC-learning to FPAC-learning, where we consider minimization of malfare rather than risk (expected loss) objectives. Both PAC and FPAC learning are parameterized by a *learning task* (model space and loss function), and we explore the resulting learnability hierarchy.
3. We show that for many loss functions, PAC and FPAC learning are *statistically equivalent*, and convexity conditions for *computationally efficient* PAC-learning are also sufficient for FPAC-learning.

## 1.1 Related Work

Constraint-based notions of algorithmic fairness have risen to prominence in fair ML, with the potential to ensure fairness (i.e., via *parity constraints* between the various groups, such as equalized odds, demographic parity, equality of opportunity, equality of outcome, etc.), thus correcting for some forms of data or algorithmic bias. While noble in intent and intuitive by design, fairness via such statistical constraints has several prominent flaws: most notably, several popular parity constraints are mutually unsatisfiable [16], and their constraint-based formulation inherently puts *accuracy* and *fairness* at odds, where additional *tolerance parameters* are required to strike a balance between the two. Furthermore, recent works [14, 15] have shown that *welfare* and even *disadvantaged group utility* can be harmed by such fairness constraints, calling into question whether these constraints actually benefit those that they purport to aid.

Perhaps in response to these issues, some recent work has trended toward welfare-based fairness-concepts, wherein both *accuracy* and *fairness* are encoded in a *welfare function* defined on a *group of subpopulations*. Welfare is then directly optimized [14, 23, 28] or constrained [13, 29] to promote *fair learning* across *all groups*. Perhaps the most similar to our work is a method of [14], wherein they *directly maximize* empirical welfare over linear classifiers; however, as with other previous works, an appropriate utility function must be selected, which we avoid by instead using malfare. We argue that *empirical welfare maximization* is an effective strategy when an appropriate and natural measure of *utility* is available, but in ML contexts, there is no "correct" or clearly neutral way to generically convert loss to utility. We avoid this issue by working directly with malfare and loss.

Our most poignant contrast to existing theoretical work is the *Seldonian learner* [30] framework, which essentially extends PAC-learning to learning problems with both *constraints* and *arbitrary nonlinear objectives*. We argue that this generality is harmful to the utility of the concept as a mathematical or practical object, as nearly any ML problem can be posed as a constrained nonlinear optimization task. The utility in FPAC learning is that it is sophisticated enough to handle fairness issues with an axiomatically-justified objective, but remains simple enough to study as a mathematical object, leading to informative reductions between various PAC and FPAC learnable classes. A similar framework, termed "multi-group agnostic PAC learning" [3, 24], also considers per-group risk values, but there *regret* (maximum relative dissatisfaction), rather than malfare, is minimized.

## 2 Aggregating Sentiment within Populations

A generic *aggregator function* $\mathrm{M}(\mathcal{S}; \boldsymbol{w})$ summarizes some *sentiment vector* $\mathcal{S} \in \mathbb{R}_{0+}^g$, across a population of $g$ groups, weighted by a *probability vector* $\boldsymbol{w} \in \mathbb{R}_+^g$ s.t. $\|\boldsymbol{w}\|_1 = 1$. When $\mathcal{S}$ measures a *desirable quantity*, generally termed *utility*, the aggregator function is a measure of *cardinal welfare* [19], and thus quantifies overall *wellbeing*. We also consider the inverse-notion, that of overall *illbeing*, termed *malfare*, in terms of an *undesirable* $\mathcal{S}$, generally *loss* or *risk*, which naturally extends the welfare concept. We show an equivalent *axiomatic justification* for malfare, and argue that its use is more natural in many situations, particularly when considering or optimizing *loss functions*.

**Definition 2.1** (Aggregator Functions: Welfare and Malfare). An *aggregator function* $\mathrm{M}(\mathcal{S}; \boldsymbol{w})$ measures the *overall sentiment* of a population, given *sentiment vector* $\mathcal{S}$ and *probability vector* $\boldsymbol{w}$. If $\mathcal{S}$ denotes a desirable quantity (e.g., utility), we call $\mathrm{M}(\mathcal{S}; \boldsymbol{w})$ a *welfare function*, written $\mathrm{W}(\mathcal{S}; \boldsymbol{w})$, and inversely, if it is undesirable (e.g., disutility, loss, or risk), we call $\mathrm{M}(\mathcal{S}; \boldsymbol{w})$ a *malfare function*, written $\mathcal{M}(\mathcal{S}; \boldsymbol{w})$.

For now, think of the term *aggregator function* as signifying that an entire population, with diverse and subjective desiderata, is considered and summarized into an *aggregate value* $\mathrm{M}(\mathcal{S}; \boldsymbol{w})$, rather than a single group's perspective (i.e., some $\mathcal{S}_i$). As we introduce axioms and show consequent properties, the appropriateness of the term shall become more apparent. We use the term *sentiment* to refer to $\mathcal{S}$ with neutral connotation, but when discussing welfare or malfare, we often refer to $\mathcal{S}$ as *utility* or *risk*, respectively, as in these cases, $\mathcal{S}$ describes a well-understood preëxisting concept. We shall see that often aggregator functions on utility values and risk values are mathematically interchangeable, however, in order to promote fairness, the desirable axioms of malfare and welfare functions differ slightly. The notation reflects the distinction; $\mathrm{M}(\mathcal{S}; \boldsymbol{w})$ is M for *mean*, whereas $\mathrm{W}(\mathcal{S}; \boldsymbol{w})$ is W for *welfare*, and $\mathcal{M}(\mathcal{S}; \boldsymbol{w})$ is $\mathcal{M}$ (*inverted* W), to emphasize its inverted nature.

### 2.1 Axioms of Cardinal Welfare and Malfare

**Definition 2.2** (Axioms of Cardinal Welfare and Malfare). We define the *aggregator-function axioms* for aggregator function $\mathrm{M}(\mathcal{S}; \boldsymbol{w})$ below. For each item, assume (if necessary) that the axiom applies $\forall \mathcal{S}, \mathcal{S}' \in \mathbb{R}_{0+}^g$, scalars $\alpha, \beta \in \mathbb{R}_{0+}$, and probability vector $\boldsymbol{w} \in \mathbb{R}_+^g$.

1. *Strict Monotonicity*: $\mathcal{S}' \neq \boldsymbol{0} \implies \mathrm{M}(\mathcal{S}; \boldsymbol{w}) < \mathrm{M}(\mathcal{S} + \mathcal{S}'; \boldsymbol{w})$.
2. *Weighted Symmetry*:[1] Suppose $\mathcal{S}' \in \mathbb{R}_{0+}^{g'}$ and probability vector $\boldsymbol{w}' \in \mathbb{R}_{0+}^{g'}$, such that for all $u \in \mathbb{R}_{0+}$ it holds $\sum_{i:\mathcal{S}_i=u} \boldsymbol{w}_i = \sum_{i:\mathcal{S}_i=u} \boldsymbol{w}_i'$. Then $\mathrm{M}(\mathcal{S}; \boldsymbol{w}) = \mathrm{M}(\mathcal{S}'; \boldsymbol{w}')$.
3. *Continuity*: $\mathrm{M}(\mathcal{S}; \boldsymbol{w})$ is a continuous function in both $\mathcal{S}$ and $\boldsymbol{w}$.
4. *Independence of Unconcerned Agents*:

$$\mathrm{M}(\langle \mathcal{S}_{1:g-1}, \alpha \rangle; \boldsymbol{w}) \leq \mathrm{M}(\langle \mathcal{S}_{1:g-1}', \alpha \rangle; \boldsymbol{w}) \implies \mathrm{M}(\langle \mathcal{S}_{1:g-1}, \beta \rangle; \boldsymbol{w}) \leq \mathrm{M}(\langle \mathcal{S}_{1:g-1}', \beta \rangle; \boldsymbol{w}) \ .$$

5. *Independence of Common Scale*: $\mathrm{M}(\mathcal{S}; \boldsymbol{w}) \leq \mathrm{M}(\mathcal{S}'; \boldsymbol{w}) \implies \mathrm{M}(\alpha \mathcal{S}; \boldsymbol{w}) \leq \mathrm{M}(\alpha \mathcal{S}'; \boldsymbol{w})$.
6. *Multiplicative Linearity*: $\mathrm{M}(\alpha \mathcal{S}; \boldsymbol{w}) = \alpha \mathrm{M}(\mathcal{S}; \boldsymbol{w})$.
7. *Unit Scale*: $\mathrm{M}(\boldsymbol{1}; \boldsymbol{w}) = \mathrm{M}(\langle 1, \ldots, 1 \rangle; \boldsymbol{w}) = 1$.
8. *Pigou-Dalton Transfer Principle*: Suppose $\mu = \boldsymbol{w} \cdot \mathcal{S} = \boldsymbol{w} \cdot \mathcal{S}'$, and for all $i \in \{1, \ldots, g\}$: $|\mu - \mathcal{S}_i'| \leq |\mu - \mathcal{S}_i|$. Then $\mathrm{W}(\mathcal{S}'; \boldsymbol{w}) \geq \mathrm{W}(\mathcal{S}; \boldsymbol{w})$.
9. *Negated Pigou-Dalton Transfer Principle*: Suppose as in 8, but conclude $\mathcal{M}(\mathcal{S}'; \boldsymbol{w}) \leq \mathcal{M}(\mathcal{S}; \boldsymbol{w})$.

We take a moment to comment on each axiom, to preview their purpose and assure the reader of their necessity. Axioms 1-5 are essentially the standard *axioms of cardinal welfarism* [22, 25]. Together, they imply (via the Debreu-Gorman theorem [10, 12]) that any aggregator function can be decomposed as a *monotonic function* of a *sum* (over groups) of *logarithm* or *power* functions.

Axiom 6 is a natural and useful property, and one which enables *dimensional analysis* on mean functions; in particular, the *units* of aggregator function match those of sentiment values. Note that axiom 6 implies axiom 5, and is thus a simple strengthening of a traditional cardinal welfare axiom. We will also see that it is essential to show convenient *statistical* and *learnability* properties. Axiom 7

---

[1]In the unweighted case, it is standard to define symmetry as simply $\mathrm{M}(\mathcal{S}) = \mathrm{M}(\pi(\mathcal{S}))$ for all permutations $\pi$ over $\{1, \ldots, g\}$, but with weightings, the simple extension $\mathrm{M}(\mathcal{S}; \boldsymbol{w}) = \mathrm{M}(\pi(\mathcal{S}); \pi(\boldsymbol{w}))$ is not quite sufficient. In particular, in this weaker form, nowhere is *additive decomposability*, wherein a group may be decomposed into multiple groups of the same sentiment value and total weight without impacting the aggregate, codified.

furthers this theme, as it ensures that not only do *units* of means match those of $\mathcal{S}$, but *scale* does as well (making comparisons like "$\mathcal{S}_i$ is *above* the population's welfare" meaningful), and also enabling comparison *across populations* (i.e., comparing *averages* rather than *sums*).

Finally, axiom 8, the *Pigou-Dalton transfer principle* [9, 21], is also standard in cardinal welfare theory, as it ensures fairness in the sense that welfare is higher when utility values are more uniform (i.e., incentivizing *equitable redistribution of "wealth"* in welfare). Its antithesis, axiom 9, encourages the opposite; in the context of welfare, this perversely incentivizes inequality, but for malfare, which we generally wish to *minimize*, the opposite occurs, thus this axiom characterizes *fairness* in the context of *malfare*. We thus state axioms 8-9 specifically for welfare or malfare, respectively, whereas axioms 1-7 apply equally well to both welfare and malfare.

Axioms 1-5, are modified from their standard presentation to include the weighting $\boldsymbol{w}$, wherein *continuity* (3) must also hold over $\boldsymbol{w}$, and *weighted symmetry* (2) specifies additionally that weight may be transferred between groups with equal sentiment value. Axioms 6-7 are novel to this work, and are key in strengthening the Debreu-Gorman theorem to ensure that all welfare and malfare functions are *power-means* in the sequel. Axiom 9 is also novel, as it is necessary to flip the inequality of axiom 8 when the sense of the aggregator function is inverted from welfare to malfare; in particular, the semantic meaning shifts from requiring that "redistribution of utility is desirable" to "redistribution of disutility is not undesirable."

**The Power-Mean**   We now define the *p-power-mean* $\mathrm{M}_p(\cdot; \cdot)$, for any $p \in \mathbb{R} \cup \pm\infty$, which we shall use to quantify both malfare and welfare. Power-means arise when analyzing aggregator functions obeying the various axioms of definition 2.2, and are thus particularly important to this work.

**Definition 2.3** (Power-Mean Welfare and Malfare). Suppose $p \in \mathbb{R} \cup \pm\infty$, sentiment vector $\mathcal{S} \in \mathbb{R}_{0+}^g$ and probability vector $\boldsymbol{w} \in \mathbb{R}_+^g$. We then define the *p-weighted-power-mean* as

$$\mathrm{M}_p(\mathcal{S}; \boldsymbol{w}) \doteq \sqrt[p]{\sum_{i=1}^g \boldsymbol{w}_i \mathcal{S}_i^p} \ , \ \mathrm{M}_0(\mathcal{S}; \boldsymbol{w}) \doteq \exp\left(\sum_{i=1}^g \boldsymbol{w}_i \ln \mathcal{S}_i\right) \ , \ \mathrm{M}_{\pm\infty}(\mathcal{S}; \boldsymbol{w}) \doteq \pm \max_{i \in 1, \ldots, g} \pm \mathcal{S}_i \ ,$$

where $p \in \{-\infty, 0, \infty\}$ attain their *limits*, the *minimum*, *geometric mean*, and *maximum*, respectively.

**Theorem 2.4** (Properties of the Power-Mean). Suppose sentiment vectors $\mathcal{S}, \mathcal{S}' \in \mathbb{R}_{0+}^g$ and probability vector $\boldsymbol{w} \in \mathbb{R}_+^g$. The following then hold.

1. *Monotonicity*: $\mathrm{M}_p(\mathcal{S}; \boldsymbol{w})$ is weakly-monotonically-increasing in $p$.
2. *Subadditivity*: $\forall p \geq 1 : \mathrm{M}_p(\mathcal{S} + \mathcal{S}'; \boldsymbol{w}) \leq \mathrm{M}_p(\mathcal{S}; \boldsymbol{w}) + \mathrm{M}_p(\mathcal{S}'; \boldsymbol{w})$.
3. *Contraction*: $\forall p \geq 1 : \left|\mathrm{M}_p(\mathcal{S}; \boldsymbol{w}) - \mathrm{M}_p(\mathcal{S}'; \boldsymbol{w})\right| \leq \mathrm{M}_p(\left|\mathcal{S} - \mathcal{S}'\right|; \boldsymbol{w}) \leq \left\|\mathcal{S} - \mathcal{S}'\right\|_\infty$.
4. *Curvature*: $\mathrm{M}_p(\mathcal{S}; \boldsymbol{w})$ is concave in $\mathcal{S}$ for $p \leq 1$ and convex in $\mathcal{S}$ for $p \geq 1$.

## 2.2   Properties of Welfare and Malfare Functions

We now derive properties of welfare and malfare from the axioms of definition 2.2.

**Theorem 2.5** (Aggregator Function Properties). Suppose aggregator function $\mathrm{M}(\mathcal{S}; \boldsymbol{w})$, and assume arbitrary sentiment vector $\mathcal{S} \in \mathbb{R}_{0+}^g$ and probability vector $\boldsymbol{w} \in \mathbb{R}_+^g$. If $\mathrm{M}(\cdot; \cdot)$ satisfies (subsets of) the aggregator-function axioms (see definition 2.2), then $\mathrm{M}(\cdot; \cdot)$ exhibits the following properties.

1. *Identity*: Axioms 6-7 imply that $\forall \alpha \in \mathbb{R}_{0+} : \mathrm{M}(\alpha \boldsymbol{1}; \boldsymbol{w}) = \alpha$.
2. *Debreu-Gorman Factorization*: Axioms 1-5 imply $\exists p \in \mathbb{R}$, strictly-monotonically-increasing continuous $F : \mathbb{R} \to \mathbb{R}_{0+}$ s.t.

$$\mathrm{M}(\mathcal{S}; \boldsymbol{w}) = F\left(\sum_{i=1}^g \boldsymbol{w}_i f_p(\mathcal{S}_i)\right) \ , \ \text{with} \ \begin{cases} p = 0 & f_0(x) \doteq \ln(x) \\ p \neq 0 & f_p(x) \doteq \mathrm{sgn}(p) x^p \end{cases} .$$

3. *Power-Mean Factorization*: Axioms 1-7 imply $F(x) = f_p^{-1}(x)$, thus $\mathrm{M}(\mathcal{S}; \boldsymbol{w}) = \mathrm{M}_p(\mathcal{S}; \boldsymbol{w})$.
4. *Fair Welfare*: Axioms 1-5 and 8 imply $p \in (-\infty, 1]$.
5. *Fair Malfare*: Axioms 1-5 and 9 imply $p \in [1, \infty)$.

Theorem 2.5 tells us that the mild conditions of axioms 1-5 (generally assumed for welfare), along with 6 (multiplicative linearity), imply that welfare and utility, or malfare and loss, are measured in

the *same units* (e.g., *nats* of *cross-entropy loss*, or *dollars* of *income utility*), and the power-mean is effectively the only reasonable family of welfare or malfare function; even without axiom 6, axioms 1-5 imply aggregator function functions are still *monotonic transformations* of power-means. Furthermore, the entirely milquetoast *unit scale* axiom (7) implies that sentiment values and aggregator functions have the same *scale*, making comparisons like "the risk of group $i$ is above (or below) the population's malfare" meaningful. Finally, we have that $W_p(\cdot; \boldsymbol{w})$ promotes equitable redistribution of utility for $p \leq 1$, and $\mathbb{M}_p(\cdot; \boldsymbol{w})$ promotes equitable redistribution of loss for $p \geq 1$.

# 3 Statistical and Computational Learning-Efficiency Guarantees

In this section, we define a formal notion of fair-learnability, termed *FPAC-learning*, where a loss function and hypothesis class are FPAC-learnable essentially if any distribution can be learned to *approximate malfare-optimality* from a *finite sample* (w.h.p.). We then construct various FPAC learners, and relate the concept to standard PAC learning [31], with the understanding that this allows the vast breadth of research on PAC-learning algorithms to be applied to FPAC learning. In particular, we show a hierarchy of fair-learnability via generic statistical and computational reductions.

**Definition 3.1** (Hypothesis Classes and Class Sequences). A *hypothesis class* is a family of functions mapping domain $\mathcal{X}$ to codomain $\mathcal{Y}$, and a *hypothesis class sequence* $\mathcal{H} = \mathcal{H}_1, \mathcal{H}_2, \ldots$ is a *nested sequence* of *hypothesis classes*, each mapping $\mathcal{X} \to \mathcal{Y}$. In other words, $\mathcal{H}_1 \subseteq \mathcal{H}_2 \subseteq \cdots$.

For example, *linear classifiers* naturally form a sequence of families using their *dimension* as $\mathcal{H}_d \doteq \left\{ \vec{x} \mapsto \mathrm{sgn}\big(\vec{x} \cdot \langle \vec{w}_{1:d}, \vec{0}\rangle\big) \mid \vec{w} \in \mathbb{R}^d \right\}$. The *hypothesis class sequence* concept allows us to distinguish statistically-easy problems, like learning hyperplanes in finite-dimensional $\mathbb{R}^d$, from statistically-challenging problems, like learning hyperplanes in $\mathbb{R}^\infty$. It is also used to analyze the *computational complexity* of learners as $d$ increases.

## 3.1 Empirical Malfare Minimization and the Rademacher Average

We define the *risk* of hypothesis $h : \mathcal{X} \to \mathcal{Y}$ w.r.t. loss $\ell : \mathcal{Y} \times \mathcal{Y} \to \mathbb{R}_+$ on distribution $\mathcal{D}$ over $(\mathcal{X} \times \mathcal{Y})$, and the *empirical risk* on sample $\boldsymbol{z} \in (\mathcal{X} \times \mathcal{Y})^m$, as

$$\mathrm{R}(h; \ell, \mathcal{D}) \doteq \mathop{\mathbb{E}}_{(x,y) \sim \mathcal{D}}\big[\ell(y, h(x))\big] \quad \& \quad \hat{\mathrm{R}}(h; \ell, \boldsymbol{z}) \doteq \mathop{\hat{\mathbb{E}}}_{(x,y) \in \boldsymbol{z}}\big[\ell(y, h(x))\big] ,$$

respectively. The goal in ML is generally to recover the $h^*$ that minimizes (true) risk, and *empirical risk minimization* (ERM) computes $\hat{h} \doteq \mathrm{argmin}_{h \in \mathcal{H}} \hat{\mathrm{R}}(h; \ell, \boldsymbol{z})$ as a proxy for $h^* \doteq \mathrm{argmin}_{h \in \mathcal{H}} \mathrm{R}(h; \ell, \mathcal{D})$. Analogously, for *fair ML*, we define *empirical malfare minimization* (EMM), given $\mathbb{M}(\cdot; \boldsymbol{w})$, $\mathcal{D}_{1:g}$, and $\boldsymbol{z}_{1:g}$, with empirical and true optimal models

$$\hat{h} \doteq \mathop{\mathrm{argmin}}_{h \in \mathcal{H}} \mathbb{M}\big(i \mapsto \hat{\mathrm{R}}(h; \ell, \boldsymbol{z}_i); \boldsymbol{w}\big) \quad \& \quad h^* \doteq \mathop{\mathrm{argmin}}_{h \in \mathcal{H}} \mathbb{M}\big(i \mapsto \mathrm{R}(h; \ell, \mathcal{D}_i); \boldsymbol{w}\big) .$$

Although empirical risk is an *unbiased estimate* of risk, with $\mathbb{M}_p(\cdot; \boldsymbol{w})$ for $p > 1$, *empirical malfare* is a *biased estimator* of malfare. Fortunately, empirical malfare is a *consistent estimator* of malfare, and we show that standard tools for bounding the error of ERM can be applied to EMM. Here it is necessary to consider not the loss function or hypothesis class *in isolation*, but their *composition*

$$\forall h \in \mathcal{H} : (\ell \circ h)(x, y) \doteq \ell(y, h(x)) \quad \& \quad \ell \circ \mathcal{H} \doteq \{\ell \circ h \mid h \in \mathcal{H}\} .$$

The *empirical Rademacher average* is a well-studied [8, 18, 26] data-dependent measurement of the *capacity to overfit* of a model class $\mathcal{H}$ w.r.t. loss function $\ell$, defined as

$$\hat{\mathfrak{R}}_m(\ell \circ \mathcal{H}, \boldsymbol{z}) \doteq \mathop{\mathbb{E}}_{\boldsymbol{\sigma}}\left[\sup_{h \in \mathcal{H}} \left| \frac{1}{m} \sum_{i=1}^m \boldsymbol{\sigma}_i \ell\big(\boldsymbol{y}_i, h(\boldsymbol{x}_i)\big)\right|\right] ,$$

where $\boldsymbol{\sigma}_{1:m}$ are drawn i.i.d. Rademacher (i.e., uniform on $\pm 1$), which essentially measures the ability of the hypothesis class to *spuriously correlate* loss values with *noise*.

**Theorem 3.2** (Generalization Guarantees for Malfare Estimation). Suppose hypothesis class $\mathcal{H} \subseteq \mathcal{X} \to \mathcal{Y}$, bounded *loss function* $\ell : (\mathcal{Y} \times \mathcal{Y}) \to [0, r]$, and per-group samples $\boldsymbol{z}_i \sim \mathcal{D}_i^m$. Then, with probability at least $1 - \delta$ over choice of $\boldsymbol{z}$, it holds *simultaneously* for all *fair malfare functions* $\mathbb{M}(\cdot; \cdot)$ (i.e., $\mathbb{M}_p(\cdot; \cdot)$ for $p \geq 1$) and *probability vectors* $\boldsymbol{w} \in \mathbb{R}_+^g$, that

$$\sup_{h \in \mathcal{H}} \left| \mathbb{M}\big(i \mapsto \mathrm{R}(h; \ell, \mathcal{D}_i); \boldsymbol{w}\big) - \mathbb{M}\big(i \mapsto \hat{\mathrm{R}}(h; \ell, \boldsymbol{z}_i); \boldsymbol{w}\big)\right| \leq \mathbb{M}\left(i \mapsto 2\hat{\mathfrak{R}}_m(\ell \circ \mathcal{H}, \boldsymbol{z}_i) + 3r\sqrt{\tfrac{\ln \frac{g}{\delta}}{2m}}; \boldsymbol{w}\right) .$$

Figure 1: We minimize malfare of a *weighted hinge-loss* SVM, with $g = 5$ ethnoracial protected groups, listed in the legend with *group weight* $w_i$ (population frequency) and *class bias* $b_i$ (proportion with income $\geq$ \$50k/yr.). Due to existing societal inequity, *class imbalance* varies widely by group, so we weight risk by $\frac{1}{b_i}$. We report per-group training (dotted) and test (dashed) hinge risks, along with the malfare objective (green) of the EMM solution

$$\hat{h} \doteq \operatorname*{argmin}_{h \in \mathcal{H}} \mathbb{M}_p\left(i \mapsto \frac{1}{b_i}\hat{\mathrm{R}}(h; \ell_{\mathrm{hinge}}, z_i); w\right) \;,$$

as functions of $p \in [1, 32]$. The experimental setup is detailed in appendix A.1.

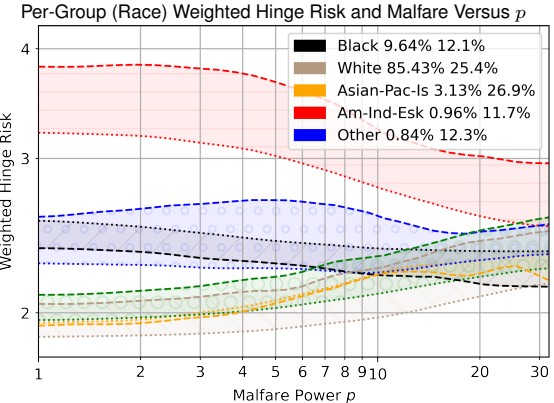

Per-Group (Race) Weighted Hinge Risk and Malfare Versus $p$

| Black 9.64% 12.1% |
| White 85.43% 25.4% |
| Asian-Pac-Is 3.13% 26.9% |
| Am-Ind-Esk 0.96% 11.7% |
| Other 0.84% 12.3% |

The *contraction property* (theorem 2.4 item 3) of malfare is key to this result, and this property does not hold for welfare. Strikingly, fair welfare functions $\mathrm{W}_p(\mathcal{S}; w)$ for $p \in [0, 1)$ are Lipschitz-discontinuous; e.g., the *Nash social welfare* (geometric welfare) $\mathrm{W}_0(\mathcal{S}; \frac{1}{g}) = \sqrt[g]{\prod_{i=1}^{g} \mathcal{S}_i}$ is unstable to perturbations of each $\mathcal{S}_i$ around 0, which causes great difficulty in sampling-based welfare estimation. For example, if utility samples are $\mathrm{BERNOULLI}(q)$-distributed for $\mathcal{S}_1$, the sample complexity of $\varepsilon$-$\delta$ estimating $\mathrm{W}_0(\mathcal{S}; w)$ grows unboundedly as $w_1 \to 0$, $q \to 0$ (jointly).

**Experimental Validation** Figure 1 presents a brief experiment on the lauded `adult` dataset, where the task is to predict whether income is above or below \$50k/yr. We train $\mathbb{M}_p(\cdot; w)$-minimizing SVM, and find significant risk-variation between groups; generally low risk for the *white* and *Asian* groups, and high risk for the *native American* and *other* groups. The $p = 1$ model is a standard weighted SVM, with poor performance for small and traditionally marginalized groups, as expected in an $85.43\%$ majority-*white* population. As $p$ increases (towards egalitarianism), we observe interesting fairness tradeoffs; training malfare increases monotonically, and in general (but not monotonically[2]), *white* and *Asian* training risks increase, as the remaining risks decrease. At first, most improvement is in the relatively-large ($9.64\%$), high-risk *Black* group, but for larger $p$, the much smaller ($0.96\%$), but higher-risk, *native American* group sharply improves.

Both training and test performance generally improve for high-risk groups, but significant overfitting occurs in small groups and malfare. This is unsurprising, since although SVM generalization error is well-understood [see 26, chapter 26], bounds are generally vacuous for tiny subpopulations of $\approx 400$ individuals. In general, overfitting increases with $p$, due to the higher relative-importance of small but high-risk groups on $\hat{h}$. This experiment validates EMM as a fair-learning technique, with the capacity to specify tradeoffs between majority and marginalized groups, while demonstrating *overfitting to fairness*, which we formally treat in the sequel. We observe similar fairness tradeoffs in our supplementary experiments (appendix A.2), on weighted and unweighted SVM and logistic regressors with *race* and *sex* groups.

## 3.2 Fair Probably Approximately Correct Learning

For context, we first present a generalized notion of PAC-learnability, which we then generalize to FPAC-learnability. Standard presentations consider only classification under 0-1 loss, but we follow the generalized learning setting of [32], which treats generic learning problems.

**Definition 3.3** (PAC-Learnability). Suppose *hypothesis class sequence* $\mathcal{H}_1 \subseteq \mathcal{H}_2 \subseteq \ldots$, all over $\mathcal{X} \to \mathcal{Y}$, and *loss function* $\ell : \mathcal{Y} \times \mathcal{Y} \to \mathbb{R}_{0+}$. We say $\mathcal{H}$ is *PAC-learnable* w.r.t. $\ell$ if there exists a (randomized) algorithm $\mathcal{A}$, such that for all

1. sequence indices $d$;
2. instance distribution $\mathcal{D}$ over $\mathcal{X} \times \mathcal{Y}$;
3. additive approximation error $\varepsilon > 0$; and
4. failure probability $\delta \in (0, 1)$;

---

[2]Note that for continuous loss functions and $g = 2$ groups, group training risks are monotonic in $p$, as seen in the supplementary *sex-based* experiments (see appendix A.2 ).

$\mathcal{A}$ can identify a hypothesis $\hat{h} \in \mathcal{H}$, i.e., $\hat{h} \leftarrow \mathcal{A}(\mathcal{D}, \varepsilon, \delta, d)$, such that

1. there exists some *sample complexity* function $m(\varepsilon, \delta, d) : (\mathbb{R}_+ \times (0,1) \times \mathbb{N}) \to \mathbb{N}$ s.t. $\mathcal{A}(\mathcal{D}, \varepsilon, \delta, d)$ consumes no more than $m(\varepsilon, \delta, d)$ samples from $\mathcal{D}$ (i.e., finite sample complexity); and
2. with probability at least $1 - \delta$ (over randomness of $\mathcal{A}$), $\hat{h}$ obeys

$$\mathrm{R}(\hat{h}; \ell, \mathcal{D}) \leq \inf_{h^* \in \mathcal{H}} \mathrm{R}(h^*; \ell, \mathcal{D}) + \varepsilon .$$

The class of such learning problems is denoted PAC, with membership denoted $(\mathcal{H}, \ell) \in \mathrm{PAC}$.

Furthermore, if for all $d$, the space of $\mathcal{D}$ is restricted such that $\inf_{h \in \mathcal{H}_d} \mathrm{R}(h; \ell, \mathcal{D}) = 0$, then $(\mathcal{H}, \ell)$ is *realizable-PAC-learnable*, written $(\mathcal{H}, \ell) \in \mathrm{PAC}^0$.

We now generalize this concept to *fair-PAC learnability*. In particular, we replace the *univariate risk-minimization* task with a *multivariate malfare-minimization* task. Following the theory of section 2.2, we do not commit to any particular objective, but instead require that a FPAC-learner is able to minimize *any* fair malfare function satisfying the standard axioms. Furthermore, here problem instances grow not just in problem complexity $d$, but also in the number of groups $g$.

**Definition 3.4** (Fair-PAC (FPAC) Learnability). Suppose *hypothesis class sequence* $\mathcal{H}_1 \subseteq \mathcal{H}_2 \subseteq \cdots \subseteq \mathcal{X} \to \mathcal{Y}$, and *loss function* $\ell : \mathcal{Y} \times \mathcal{Y} \to \mathbb{R}_{0+}$. We say $\mathcal{H}$ is *fair-PAC-learnable* w.r.t. $\ell$ if there exists a (randomized) algorithm $\mathcal{A}$, such that for all

1. sequence indices $d$;
2. $g$ instance distributions $\mathcal{D}_{1:g}$ over $(\mathcal{X} \times \mathcal{Y})^g$;
3. probability vectors $\boldsymbol{w} \in \mathbb{R}_+^g$;
4. malfare concepts $\mathbb{M}$ satisfying axioms 1-7+9;
5. additive approximation errors $\varepsilon > 0$; and
6. failure probabilities $\delta \in (0,1)$;

$\mathcal{A}$ can identify a hypothesis $\hat{h} \in \mathcal{H}$, i.e., $\hat{h} \leftarrow \mathcal{A}(\mathcal{D}_{1:g}, \boldsymbol{w}, \mathbb{M}, \varepsilon, \delta, d)$, such that
1. there exists some *sample complexity* function $m(\varepsilon, \delta, d, g) : (\mathbb{R}_+ \times (0,1) \times \mathbb{N} \times \mathbb{N}) \to \mathbb{N}$ s.t. $\mathcal{A}(\mathcal{D}_{1:g}, \boldsymbol{w}, \mathbb{M}, \varepsilon, \delta, d)$ consumes no more than $m(\varepsilon, \delta, d, g)$ samples (finite sample complexity); and
2. with probability at least $1 - \delta$ (over randomness of $\mathcal{A}$), $\hat{h}$ obeys

$$\mathbb{M}\big(i \mapsto \mathrm{R}(\hat{h}; \ell, \mathcal{D}_i); \boldsymbol{w}\big) \leq \inf_{h^* \in \mathcal{H}} \mathbb{M}\big(i \mapsto \mathrm{R}(h^*; \ell, \mathcal{D}_i); \boldsymbol{w}\big) + \varepsilon .$$

The class of such fair-learning problems is FPAC, with membership denoted $(\mathcal{H}, \ell) \in \mathrm{FPAC}$.

Finally, if for all $d$, the space of $\mathcal{D}$ is restricted such that $\inf_{h \in \mathcal{H}_d} \max_{i \in 1, \dots, g} \mathrm{R}(h; \ell, \mathcal{D}_i) = 0$, then $(\mathcal{H}, \ell)$ is *realizable-FPAC-learnable*, written $(\mathcal{H}, \ell) \in \mathrm{FPAC}^0$.

**Observation 3.5** (Malfare Functions and Special Cases). By assumption, $\mathbb{M}(\cdot; \cdot)$ must be $\mathbb{M}_p(\cdot; \cdot)$ for some $p \in [1, \infty)$. Taking $g = 1$ implies $\boldsymbol{w} = \langle 1 \rangle$, and $\mathbb{M}_p(\mathcal{S}; \boldsymbol{w}) = \mathcal{S}_1$, thus reducing the problem to standard PAC-learning. Similarly, taking $p = 1$ converts the problem to *weighted loss minimization* (weights determined by $\boldsymbol{w}$), and $p = \infty$ yields a *minimax optimization problem*, where the max is over groups, as commonly encountered in adversarial and robust learning settings.

**On Computational Efficiency**  Some authors consider not just the *statistical* but also the *computational* performance of learning, generally requiring that $\mathcal{A}$ have *polynomial* time complexity (and thus implicitly sample complexity). In other words, they require that $\mathcal{A}(\mathcal{D}, \varepsilon, \delta, d)$ terminates in $m(\varepsilon, \delta, d) \in \mathrm{Poly}(\frac{1}{\varepsilon}, \frac{1}{\delta}, d)$ steps. A similar concept of *polynomial-time* FPAC-learnability is equally interesting, where here we assume $\mathcal{A}(\mathcal{D}_{1:g}, \boldsymbol{w}, \mathbb{M}, \varepsilon, \delta, d)$ may be computed by a Turing machine (with access to *sampling* and *entropy* oracles) in $m(\varepsilon, \delta, d, g) \in \mathrm{Poly}(\frac{1}{\varepsilon}, \frac{1}{\delta}, d, g)$ steps. We denote these concepts $\mathrm{PAC}_{\mathrm{Poly}}$, $\mathrm{PAC}_{\mathrm{Poly}}^0$, $\mathrm{FPAC}_{\mathrm{Poly}}$, and $\mathrm{FPAC}_{\mathrm{Poly}}^0$.

**Observation 3.6** (FPAC-Learnability and Weighted Loss Functions). Per-group weighted-loss functions are a well-studied object in the fairness literature, representing the idea that different types of incorrect or undesirable outcome can impact different groups in different ways.[3] It is not immediately obvious that the FPAC framework covers this case, however observe that it is compatible with weighted loss functions, where the weighting is associated with *each individual sample*, as this is just a particular choice of loss function. This is actually a much more general case, but since FPAC-learnability requires uniform learnability *over all distributions*, this also includes distributions where per-group weightings are constant.

---

[3]The experiments of section 3.1 represent a special case of this, wherein incorrect classifications disproportionately impact marginalized groups, who already have disproportionately many low-income members.

We also observe (immediately from definitions 3.3 and 3.4) that PAC-learning is a special case of FPAC-learning; in particular, taking $g = 1$ implies $\mathrm{M}_p(\mathcal{S}) = \mathrm{M}_1(\mathcal{S}) = \mathcal{S}_1$, thus *malfare-minimization* coincides with *risk minimization*. The more interesting question, which we seek to answer in the remainder of this document, is *when* and *whether* the *converse* holds. Furthermore, when possible, we would like to show practical *constructive reductions*.

We now show that in the *realizable case*, *PAC-learnability* implies *FPAC-learnability* via a constructive polynomial-time reduction. Our reduction simply takes a sufficiently number of samples from the *uniform mixture distribution* over all $g$ groups, and PAC-learns on this distribution. As the reduction is constructive (and efficiency-preserving), this gives us generic algorithms for (efficient) realizable FPAC-learning in terms of algorithms for realizable PAC-learning.

**Theorem 3.7** (Realizable Reductions). Suppose loss function $\ell$ and hypothesis class $\mathcal{H}$. Then

1. $(\mathcal{H}, \ell) \in \mathrm{PAC}^0 \implies (\mathcal{H}, \ell) \in \mathrm{FPAC}^0$; and     2. $(\mathcal{H}, \ell) \in \mathrm{PAC}^0_{\mathrm{Poly}} \implies (\mathcal{H}, \ell) \in \mathrm{FPAC}^0_{\mathrm{Poly}}$.

We construct a(n) (efficient) FPAC-learner for $(\mathcal{H}, \ell)$ by noting that there exists some $\mathcal{A}'$ with sample-complexity $\mathrm{m}_{\mathcal{A}'}(\varepsilon, \delta, d)$ and time complexity $\mathrm{t}_{\mathcal{A}'}(\varepsilon, \delta, d)$ to PAC-learn $(\mathcal{H}, \ell)$, and taking $\mathcal{A}(\mathcal{D}_{1:g}, \boldsymbol{w}, \Lambda, \varepsilon, \delta, d) \doteq \mathcal{A}'(\mathrm{mix}(\mathcal{D}_{1:g}), \frac{\varepsilon}{g}, \delta, d)$, where $\mathrm{mix}(\mathcal{D}_{1:g})$ denotes the uniformly-weighted mixture of distributions $\mathcal{D}_{1:g}$. Then $\mathcal{A}$ FPAC-learns $(\mathcal{H}, \ell)$, with sample-complexity $\mathrm{m}_{\mathcal{A}}(\varepsilon, \delta, d, g) = \mathrm{m}_{\mathcal{A}'}(\frac{\varepsilon}{g}, \delta, d)$, and time-complexity $\mathrm{t}_{\mathcal{A}}(\varepsilon, \delta, d, g) = \mathrm{t}_{\mathcal{A}'}(\frac{\varepsilon}{g}, \delta, d)$.

Unfortunately, the argument here strongly depends on the properties of realizability, and does not extend to the agnostic case. Furthermore, we note that, philosophically speaking, realizable FPAC learning is rather uninteresting, essentially because in a world where all parties may be satisfied completely, the obvious solution is to do so. Thus nontrivial unfairness and bias issues logically only arise in a world of *conflict*, e.g., in zero-sum or resource-constrained settings, which foster *competition* between groups. We henceforth focus our efforts on the more interesting agnostic-learning setting.

### 3.3    Characterizing Fair Statistical Learnability with FPAC-Learners

We first consider only questions of *statistical learning*, i.e., we ignore computation and show that *there exist* FPAC-learning algorithms. In particular, we show a generalization of the *fundamental theorem of statistical learning* to fair learning problems. The aforementioned result relates *uniform convergence* and *PAC-learnability*, and is generally stated for binary classification only. We define a natural generalization of uniform convergence to arbitrary learning problems within our framework, and then show conditions under which a generalized fundamental theorem of (fair) statistical learning holds. In particular, we show that, neglecting computational concerns, PAC-learnability and FPAC learnability are equivalent for learning problems with a particular *no-free-lunch* guarantee.

We first define a *generalized notion* of *uniform convergence*. In particular, our definition applies to *any bounded loss function*, thus greatly generalizes the standard notion for binary classification.

**Definition 3.8** (Uniform Convergence). Suppose $\ell : \mathcal{Y} \times \mathcal{Y} \to [0, r]$ and hypothesis class $\mathcal{H} \subseteq \mathcal{X} \to \mathcal{Y}$. We say $(\mathcal{H}, \ell) \in \mathrm{UC}$ if

$$\lim_{m \to \infty} \sup_{\mathcal{D} \text{ over } \mathcal{X} \times \mathcal{Y}} \mathbb{E}_{\boldsymbol{z} \sim \mathcal{D}^m} \left[ \sup_{h \in \mathcal{H}} \left| \hat{\mathrm{R}}(h; \ell, \boldsymbol{z}) - \mathrm{R}(h; \ell, \mathcal{D}) \right| \right] = 0 \ .$$

We stress that this definition is both uniform over $\ell$ composed with the *hypothesis class* $\mathcal{H}$ and uniform over *all possible distributions* $\mathcal{D}$. Standard uniform convergence definitions consider only the convergence of *empirical frequencies* of events to their *true frequencies*, whereas we generalize to consider uniform convergence of the *empirical means* of functions to their *expected values*. It is also helpful to consider the *sample complexity* of $\varepsilon$-$\delta$ uniform-convergence, where we take

$$\mathrm{m}_{\mathrm{UC}}(\ell \circ \mathcal{H}, \varepsilon, \delta) \doteq \operatorname{argmin} \left\{ m \ \Big| \ \sup_{\mathcal{D} \text{ over } \mathcal{X} \times \mathcal{Y}} \mathbb{P} \Big( \sup_{h \in \mathcal{H}} \Big| \mathbb{E}_{\mathcal{D}}[\ell \circ h] - \hat{\mathbb{E}}_{\boldsymbol{z} \sim \mathcal{D}^m}[\ell \circ h] \Big| > \varepsilon \Big) \leq \delta \right\} \ ,$$

i.e., the *minimum sufficient sample size* to ensure $\varepsilon$-$\delta$ uniform-convergence over the *loss family* $\ell \circ \mathcal{H}$. For classification, *uniform convergence* bijectively implies *PAC-learnability* as follows.

**Theorem 3.9** (Fundamental Theorem of Statistical Learning [26, theorem 6.2]). Suppose $\ell$ is the 0-1 loss. Then the following are equivalent:

1. $\forall d \in \mathbb{N}$: $\mathcal{H}_d$ has finite Vapnik-Chervonenkis (VC) dimension [33, 34].

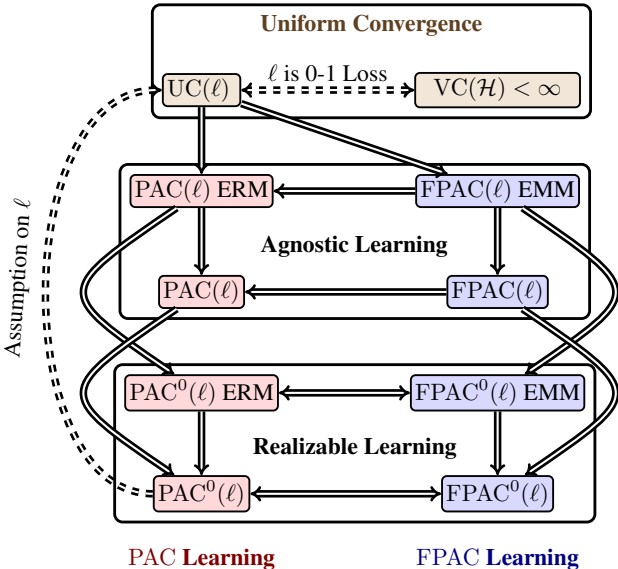

Figure 2: Implications between membership in PAC and FPAC classes. For arbitrary fixed $\ell$, $\Longrightarrow$ denotes *implication of membership* of some $\mathcal{H}$ (i.e., containment); see theorems 3.7 and 3.10, and $=\!\!\Rightarrow$ holds conditionally on $\ell$. Note that when the assumption on $\ell$ (see theorem 3.10) holds, the hierarchy collapses, and in general, under realizability, some classes are known to coincide.

2. $\forall d \in \mathbb{N}$: $(\ell, \mathcal{H}_d)$ has the uniform convergence property.
3. Any ERM rule is a successful agnostic-PAC learner for $\mathcal{H}$, thus $\mathcal{H}$ is agnostic-PAC learnable.
4. Any ERM rule is a successful realizable-PAC learner for $\mathcal{H}$, thus $\mathcal{H}$ is realizable-PAC learnable.

It is rather subtle to generalize this result beyond classification, as there are PAC-learnable problems for which uniform convergence *does not hold*. However, [1] show similar results for various regression problems, and we now generalize to FPAC learning (summarized in figure 2).

**Theorem 3.10** (Fundamental Theorem of Fair Statistical Learning). Suppose $\ell$ such that $\forall \mathcal{H} :$ $(\mathcal{H}, \ell) \in \text{PAC}^0 \implies (\mathcal{H}, \ell) \in \text{UC}$. Then $\forall \mathcal{H}$, the following are equivalent:

1. $\forall d \in \mathbb{N}$: $(\ell, \mathcal{H}_d)$ has the generalized uniform convergence property.
2. Any EMM rule is an agnostic-FPAC learner for $(\ell, \mathcal{H})$, thus $(\ell, \mathcal{H})$ is agnostic-FPAC learnable.
3. Any EMM rule is a realizable-FPAC learner for $(\ell, \mathcal{H})$, thus $(\ell, \mathcal{H})$ is realizable-FPAC learnable.

### 3.4 Characterizing Computational Learnability with Efficient FPAC Learners

In this section, we consider the more granular question of whether FPAC learning is *computationally harder* than PAC learning. In other words, where previously we showed conditions under which PAC = FPAC, here we focus on the subset of models with polynomial time training-efficiency guarantees, i.e., $\text{PAC}_{\text{Poly}} = \text{FPAC}_{\text{Poly}}$. Theorem 3.7 has already characterized the computational complexity of *realizable* FPAC-learning, so we now focus on the agnostic case. Here we show neither a generic reduction, nor a non-constructive proof that $\text{PAC}_{\text{Poly}} = \text{FPAC}_{\text{Poly}}$, nor do we show a counterexample, but rather we show that under standard convexity conditions often leveraged as sufficient for efficient PAC-learning, so too is efficient FPAC-learning possible.

Here we show concretely and constructively the existence of FPAC-learners under standard convex optimization assumptions via the *projected subgradient method* [27]. This result is immediately practical, and can be applied verbatim to problems like *generalized linear models* [20] and many *kernel methods*.

**Theorem 3.11** (Efficient FPAC Learning via Convex Optimization). Suppose each hypothesis space $\mathcal{H}_d \in \mathcal{H}$ is indexed by $\Theta_d \subseteq \mathbb{R}^{\text{Poly}(d)}$, i.e., $\mathcal{H}_d = \{h(\cdot; \theta) \mid \theta \in \Theta_d\}$, s.t. (Euclidean) $\text{Diam}(\Theta_d) \in \text{Poly}(d)$, and $\forall x \in \mathcal{X}, \theta \in \Theta_d$, $h(x; \theta)$ can be evaluated in $\text{Poly}(d)$ time, and $\tilde{\theta} \in \mathbb{R}^{\text{Poly}(d)}$ can be Euclidean-projected onto $\Theta_d$ in $\text{Poly}(d)$ time. Suppose also $\ell$ such that $\forall x \in \mathcal{X}, y \in \mathcal{Y} :$ $\theta \mapsto \ell(y, h(x; \theta))$ is a *convex function*, and suppose Lipschitz constants $\lambda_\ell, \lambda_\mathcal{H} \in \text{Poly}(d)$ and some norm $\|\cdot\|_\mathcal{Y}$ over $\mathcal{Y}$ s.t. $\ell$ is $\lambda_\ell \|\cdot\|_\mathcal{Y}$-$|\cdot|$-Lipschitz in $\hat{y}$, i.e.,

$$\forall y, \hat{y}, \hat{y}' \in \mathcal{Y} : \left|\ell(y, \hat{y}) - \ell(y, \hat{y}')\right| \leq \lambda_\ell \left\|\hat{y} - \hat{y}'\right\|_\mathcal{Y} ,$$

---

**Algorithm 1** Approximate Empirical Malfare Minimization via the Projected Subgradient Method

---

1: **procedure** $\mathcal{A}_{\mathrm{PSG}}(\ell, \mathcal{H}, \theta_0, \mathrm{m}_{\mathrm{UC}}(\cdot, \cdot), \mathcal{D}_{1:g}, \boldsymbol{w}, \mathcal{M}(\cdot; \cdot), \varepsilon, \delta)$
2:     **input**: $\lambda_\ell$-Lipschitz loss function $\ell$, $\lambda_{\mathcal{H}}$-Lipschitz hypothesis class $\mathcal{H}$ with parameter space $\Theta$ s.t. $\ell \circ \mathcal{H}$ is convex, initial guess $\theta_0 \in \Theta$, uniform-convergence sample-complexity bound $\mathrm{m}_{\mathrm{UC}}(\cdot, \cdot)$, group distributions $\mathcal{D}_{1:g}$, group weights $\boldsymbol{w} \in \mathbb{R}_+^g$, malfare function $\mathcal{M}(\cdot; \cdot)$, and additive error guarantee $\varepsilon$-$\delta$
3:     **output**: $\varepsilon$-$\delta$-$\mathcal{M}(\cdot; \cdot)$-optimal $\hat{h} \in \mathcal{H}$
4:     $\mathrm{m}_{\mathcal{A}} \leftarrow \mathrm{m}_{\mathrm{UC}}(\frac{\varepsilon}{3}, \frac{\delta}{g})$; $\boldsymbol{z}_{1:g,1:\mathrm{m}_{\mathcal{A}}} \sim \mathcal{D}_1^{\mathrm{m}_{\mathcal{A}}} \times \cdots \times \mathcal{D}_g^{\mathrm{m}_{\mathcal{A}}}$  ▷ Draw sufficient sample for each group
5:     $n \leftarrow 1 + \left( \frac{3 \operatorname{Diam}(\Theta) \lambda_\ell \lambda_{\mathcal{H}}}{\varepsilon} \right)^2$; $\alpha \leftarrow \frac{\operatorname{Diam}(\Theta)}{\lambda_\ell \lambda_{\mathcal{H}} \sqrt{n}}$         ▷ Set iteration count $n$ and learning rate $\alpha$
6:     $f(\theta) : \Theta \mapsto \mathbb{R}_{0+} \doteq \mathcal{M}\big( i \mapsto \hat{\mathrm{R}}(h(\cdot; \theta); \ell, \boldsymbol{z}_i); \boldsymbol{w} \big)$         ▷ Define empirical malfare objective
7:     $\hat{\theta} \leftarrow \textsc{ProjectedSubgradient}(f, \Theta, \theta_0, n, \alpha)$   ▷ Run PSG algorithm on empirical malfare
8:     **return** $h(\cdot; \hat{\theta})$                                                       ▷ Return $\varepsilon$-$\delta$ optimal model
9: **end procedure**

---

and also that each $\mathcal{H}_d$ is $\lambda_{\mathcal{H}}$-$\|\cdot\|_2$-$\|\cdot\|_{\mathcal{Y}}$-Lipschitz in $\theta$, i.e.,

$$\forall x \in \mathcal{X}, \theta, \theta' \in \Theta_d : \big\| h(x; \theta) - h(x; \theta') \big\|_{\mathcal{Y}} \leq \lambda_{\mathcal{H}} \big\| \theta - \theta' \big\|_2 \ .$$

Finally, assume $\ell \circ \mathcal{H}_d$ exhibits $\varepsilon$-$\delta$ *uniform convergence* with sample complexity $\mathrm{m}_{\mathrm{UC}}(\varepsilon, \delta, d) \in \operatorname{Poly}(\frac{1}{\varepsilon}, \frac{1}{\delta}, d)$. It then holds that, for arbitrary initial guess $\theta_0 \in \Theta_d$, for any group distributions $\mathcal{D}_{1:g}$, group weights $\boldsymbol{w}$, and fair malfare function $\mathcal{M}(\cdot; \cdot)$, the algorithm (see algorithm 1)

$$\mathcal{A}(\mathcal{D}_{1:g}, \boldsymbol{w}, \mathcal{M}(\cdot; \cdot), \varepsilon, \delta, d) \doteq \mathcal{A}_{\mathrm{PSG}}\big( \ell, \mathcal{H}_d, \theta_0, \mathrm{m}_{\mathrm{UC}}(\cdot, \cdot, d), \mathcal{D}_{1:g}, \boldsymbol{w}, \mathcal{M}(\cdot; \cdot), \varepsilon, \delta \big)$$

FPAC-learns $(\mathcal{H}, \ell)$ with sample complexity $\mathrm{m}(\varepsilon, \delta, d, g) = g \cdot \mathrm{m}_{\mathrm{UC}}(\frac{\varepsilon}{3}, \frac{\delta}{g}, d)$, and (training) time-complexity $\in \operatorname{Poly}(\frac{1}{\varepsilon}, \frac{1}{\delta}, d, g)$, thus $(\mathcal{H}, \ell) \in \mathrm{FPAC}_{\mathrm{Poly}}$.

## 4 Conclusion

This paper introduces *malfare minimization* as a fair learning task, which we argue is better aligned to address ML tasks cast as loss minimization than is welfare. We then show statistical and computational relationships between malfare and risk minimization. We do not claim that malfare is a better or more useful concept than welfare; rather only that it is *significantly different*, enjoys equivalent axiomatic footing, and it stands to reason that the right tool (malfare) should be used for the task (fair risk minimization). We acknowledge that some learning tasks, e.g., bandit problems and reinforcement learning tasks, are more naturally phrased as *maximizing* utility or (discounted) reward, however, most supervised learning problems are naturally cast as minimizing nonnegative *loss functions*.

We are highly interested in exploring a parallel theory of *fair welfare maximization*, however some key malfare properties do not hold for welfare. In particular, non-Lipschitz welfare functions, i.e., $\mathrm{W}_p(\cdot; \cdot)$ for $p \in [0, 1)$, create great *statistical* and *computational* challenges in learning. For this reason, straightforward translation of our FPAC framework into a welfare setting is rather vacuous. In contrast, the *contraction property* (theorem 2.4 item 3) of malfare leads to *uniform sample complexity bounds*, which are crucial to FPAC-learning. It thus seems that similar bounds for welfare would need either to either impose additional assumptions to avoid non-Lipschitz behavior (e.g., artificially limit the permitted range of $p$), or otherwise provide weaker (non-uniform) learning guarantees.

As FPAC learning generalizes PAC learning, known hardness reductions and lower-bounds apply, thus the interesting computation-theoretic question is whether *malfare minimization* is harder than *risk minimization*. Theorem 3.7 answers this question in the negative *under realizability*, as does theorem 3.10 for *sample complexity* in the general (agnostic) case, under appropriate conditions on the loss function. The remaining cases are left open, though section 3.4 shows that convexity conditions sufficient for efficient PAC-learnability are also sufficient for efficient FPAC-learnability. We hope that deeper inquiry into these questions will lead to both a better understanding of what is and is not FPAC-learnable, as well as more practical and efficient reductions and FPAC-learners.

## Acknowledgments

The author graciously acknowledges NSF grant RI-1813444 and DARPA/AFRL grant FA8750 for funding this work in part. The author also wishes to thank Cristina Menghini for her assistance with the experimental code, and Eli Upfal for insightful discussion regarding the intricacies of a generalized PAC-learnability concept.

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
