# A    Experimental Setup and Extensions

## A.1    Data, Preprocessing, and Experimental Setup

All experiments are conducted on the `adult` dataset, derived from the 1994 US Census database, and obtained from the UCI repository [11], where it was donated by Ronny Kohavi and Barry Becker. This dataset has $m = 48842$ instances, and we used a 90%:10% training:test split. The task has binary target variable `income`, 6 numeric features, and 8 categorical features, including `race` split into 5 ethnoracial groups, namely {`White`, `Black`, `Asian-Pac-Islander`, `Amer-Indian-Eskimo`(sic), `Other`}, and `sex` split into {`male`, `female`}. In each experiment, the target and protected group are omitted from the feature set, the remaining categorical features are 1-hot encoded, and all $d$ features are $z$-score normalized.

All experiments are with $\lambda$-$\ell_2$-norm constrained linear predictors, i.e., the hypothesis class is

$$\mathcal{H} \doteq \left\{ h(\vec{x}; \vec{\theta}) = \vec{x} \cdot \vec{\theta} \,\middle|\, \vec{\theta} \in \mathbb{R}^d, \|\vec{\theta}\|_2 \leq \lambda \right\} \ .$$

The output of this hypothesis class is real-valued, but for this binary classification task, we take $\mathcal{Y} = \pm 1$, so the loss function is selected to reify this value with a semantic classification interpretation. The 0-1 loss (for hard classification) is defined as

$$\ell_{01}(y, h(\vec{x}; \vec{\theta})) = 1 - y \operatorname{sgn}(\vec{x} \cdot \vec{\theta}) \ ,$$

which is readily interpreted in a decision-theoretic sense, but is generally computationally intractable to optimize. The SVM objective is generally stated in terms of the *hinge loss*, which acts as a convex relaxation of the 0-1 loss. The hinge-loss is defined as

$$\ell_{\text{hinge}}(y, h(\vec{x}; \vec{\theta})) = \max(0, 1 - y(\vec{x} \cdot \vec{\theta})) \ ,$$

which is of course *convex* in $\vec{\theta}$, and obeys $\ell_{01}(y, h(\vec{x}; \vec{\theta})) \leq \ell_{\text{hinge}}(y, h(\vec{x}; \vec{\theta}))$. Finally, the logistic-regression cross-entropy loss (measured in nats) is (see, e.g., ch. 9.3 of [26])

$$\ell_{\text{LRCE}}(y, h(\vec{x}; \vec{\theta})) = \ln\bigl(1 + \exp\bigl(-y(\vec{x} \cdot \vec{\theta})\bigr)\bigr) \ ,$$

which interprets the model output as a *probabilistic classification* $\mathbb{P}(y = 1|\hat{y}) = \frac{1}{1+\exp(-\hat{y})}$. Note that for $\hat{y} \not\approx 0$, $\ell_{\text{LRCE}}(y, \hat{y}) \approx \ell_{\text{hinge}}(y, \hat{y})$, and logistic regression may also be viewed as a *convex relaxation* of hard classification, as $\ell_{01}(y, \hat{y}) \leq \frac{1}{\ln(2)} \ell_{\text{LRCE}}$ (perhaps more naturally, the $\frac{1}{\ln(2)}$ constant vanishes if we measure cross entropy in *bits* rather than *nats*).

In all experiments with *weighted risk values*, we use regularity constraint $\lambda = 4$, and in the experiments with *unweighted risk values*, we take $\lambda = 10$.

**Implementation and Computational Resources**    Computation was not a concern on these simple convex linear models; all experiments were run on a low-end laptop with no GPU acceleration.

Theorem 3.11 analytically quantifies the computational complexity of $\varepsilon$-EMM, but in our experiments, we simply used standard out-of-the-box first-order methods (adaptive projected gradient descent and SLSQP), as well as derivative-free methods (COBYLA) to train all models.

## A.2    Supplementary Experiments

**0-1 Risk of Weighted SVM**    Figure A1 complements figure 1, reporting the same per-group and malfare statistics, except now on the (similarly weighted) 0-1 risk, rather than the weighted hinge risk. Here, the interpretation is that the hinge risk is a *convex proxy* for the 0-1 risk, as it would be computationally intractable to optimize the 0-1 risk directly. Because we optimize hinge risk, but report 0-1 risk, we don't expect to see monotonicity in malfare, and the discontinuity of the 0-1 risk is manifest as noise in risk values. Nevertheless, if hinge risk is a good proxy for 0-1 risk, we should still see a *general trend* of the classifier becoming fairer (improving high-risk group performance) w.r.t. 0-1 risk as it becomes fairer w.r.t. hinge risk, and we do in fact observe this with increasing $p$.

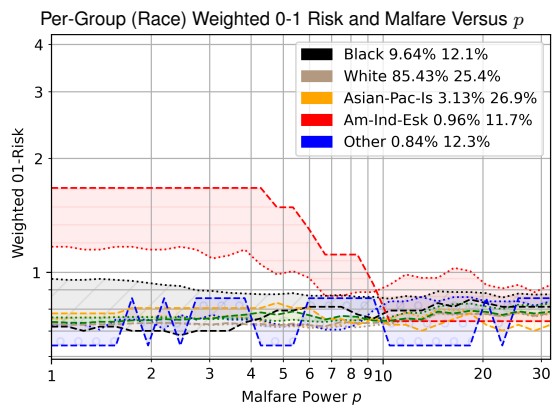

Figure A1: Training (dashed) and test (dotted) weighted 0-1 risk, per group, of hinge-risk SVM in the `adult` experiment, as functions of malfare power $p$. Training and test malfare are also plotted, again as functions of $p$. The model is optimized for weighted malfare of weighted hinge-risk, and is thus identical to the model reported on in figure 1. Here hinge-risk is optimized as a *proxy* for the 0-1 risk, so only the reported risk function changes in this figure.

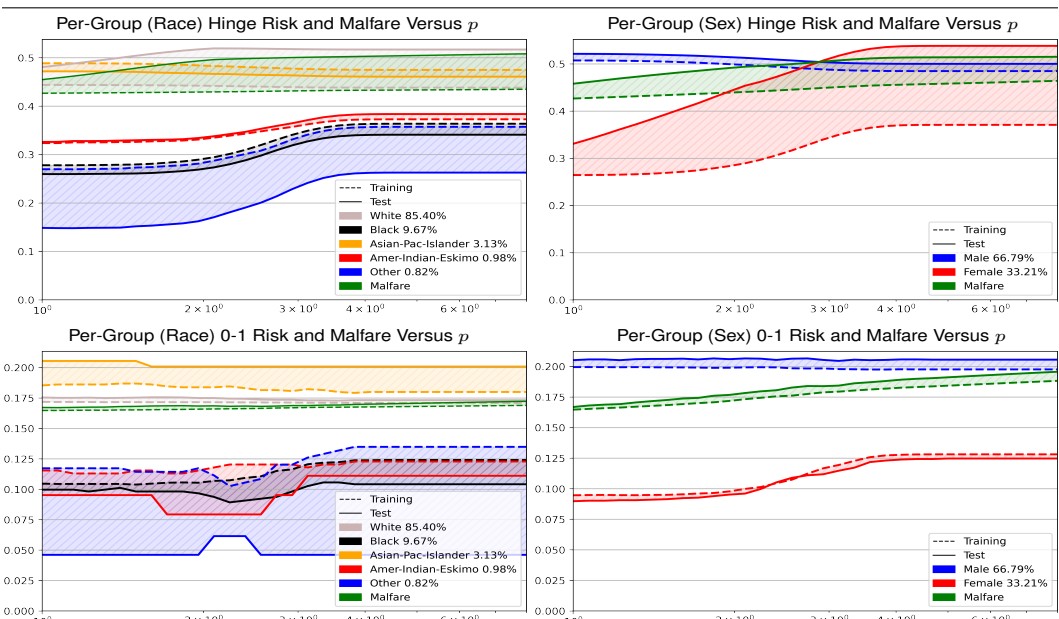

Figure A2: Unweighted linear SVM experiments on `adult` dataset, with groups split by *race* (left) and *sex* (right), malfare and risk plotted against $p$. The upper row depicts hinge-risks and malfare of hinge-risks, and the lower row depicts the 0-1 risks and malfare of 0-1 risks (of the models trained on hinge-risk). All plots show training (dashed) and test (solid) per-group risk and malfare values, as functions of $p$, with shaded regions depicting train-test gaps.

**Unweighted SVM** These experiments are quite similar to those of figure 1 and figure A1, except here we optimize the malfare of, and report the values of, the *unweighted* hinge risk. In these experiments, we also take regularity constraint $\|\vec{\theta}\|_2 \leq \lambda = 10$, and report the hinge and 0-1 risks and malfares, using *race* and *sex* groups. As such, the objective is to minimize the $\mathbb{M}_p(\cdot; \boldsymbol{w})$ malfare of per-group hinge risks, using per-group-frequencies as malfare weights, i.e.,

$$\hat{h} \doteq \operatorname*{argmin}_{h \in \mathcal{H}} \mathbb{M}_p \left( i \mapsto \hat{\mathrm{R}}(h; \ell_{\mathrm{hinge}}, \boldsymbol{z}_i); \boldsymbol{w} \right) \quad.$$

With both *sex* and *race*, we see significantly variations in model performance between groups. We stress that group size and affluence are not directly correlated with model accuracy; for instance, here we see that model performance on the (generally affluent) *Male*, *white*, and *Asian* populations is relatively poor, due to greater income homogeneity within these groups (in direct contrast to the *weighted* experiments).

In all cases, we see that increasing $p$ improves the *training set performance* of the model on the high-risk (inaccurate) groups (male, white, and Asian), at the cost of significant performance degradation

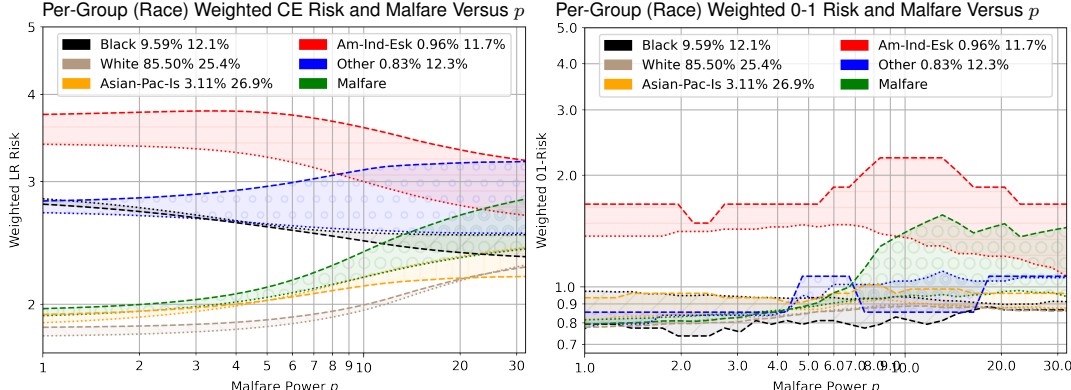

Figure A3: Experiments on the `adult` dataset, with ethnoracial protected groups, and the *weighted logistic regression* malfare objective. Training (dashed) and test (dotted) malfare and risk values are plotted as functions of $p$, with both weighted cross entropy risk (left) and weighted 0-1 risk (right).

for the more accurate groups. However, the trend does not always hold in *test set* performance, since raising $p$ increases the relative importance of *high-risk subpopulations* in training, which leads to increased overfitting. This highlights the phenomenon of *overfitting to fairness*, as we see that improved training set malfare does not necessarily translate to the test set.

**Logistic Regression Experiments**   Figure A3 complements the previous experiments, where now we optimize malfare of (weighted) *cross entropy risk* of logit predictors, where weights are chosen as in figure 1, i.e., we optimize

$$\hat{h} \doteq \underset{h \in \mathcal{H}}{\arg\min} \mathbb{M}_p \left( i \mapsto \tfrac{1}{\boldsymbol{b}_i} \hat{\mathrm{R}}(h; \ell_{\mathrm{LRCE}}, \boldsymbol{z}_i); \boldsymbol{w} \right) \quad .$$

We draw essentially the same conclusions as with the hinge risk: malfare minimization yields better training performance for high-risk (Black, native American, and other) groups, and better test performance as well, except in the *other* group, which is very small, and badly overfit.

# B   A Compendium of Missing Proofs

Here we present all missing proofs of results stated in the main text.

## B.1   Welfare and Malfare

We now show theorem 2.4.

**Theorem 2.4** (Properties of the Power-Mean). Suppose sentiment vectors $\mathcal{S}, \mathcal{S}' \in \mathbb{R}_{0+}^g$ and probability vector $\boldsymbol{w} \in \mathbb{R}_+^g$. The following then hold.

1. *Monotonicity*: $\mathrm{M}_p(\mathcal{S}; \boldsymbol{w})$ is weakly-monotonically-increasing in $p$.
2. *Subadditivity*: $\forall p \geq 1: \mathrm{M}_p(\mathcal{S} + \mathcal{S}'; \boldsymbol{w}) \leq \mathrm{M}_p(\mathcal{S}; \boldsymbol{w}) + \mathrm{M}_p(\mathcal{S}'; \boldsymbol{w})$.
3. *Contraction*: $\forall p \geq 1: \left| \mathrm{M}_p(\mathcal{S}; \boldsymbol{w}) - \mathrm{M}_p(\mathcal{S}'; \boldsymbol{w}) \right| \leq \mathrm{M}_p(\left| \mathcal{S} - \mathcal{S}' \right|; \boldsymbol{w}) \leq \left\| \mathcal{S} - \mathcal{S}' \right\|_\infty$.
4. *Curvature*: $\mathrm{M}_p(\mathcal{S}; \boldsymbol{w})$ is concave in $\mathcal{S}$ for $p \leq 1$ and convex in $\mathcal{S}$ for $p \geq 1$.

*Proof.* We omit proof of item 1, as this is a standard property of power-means, generally termed the *power-mean inequality* [4, chapter 3].

We first show item 2. By the triangle inequality (for $p \geq 1$), we have

$$\mathrm{M}_p(\mathcal{S} + \mathcal{S}'; \boldsymbol{w}) \leq \mathrm{M}_p(\mathcal{S}; \boldsymbol{w}) + \mathrm{M}_p(\mathcal{S}'; \boldsymbol{w}) \ .$$

We now show item 3 First take $\boldsymbol{\varepsilon} \doteq \mathcal{S} - \mathcal{S}'$, and let $\boldsymbol{\varepsilon}_+ \doteq \boldsymbol{0} \vee \boldsymbol{\varepsilon}$, where $\boldsymbol{a} \vee \boldsymbol{b}$ denotes the (elementwise) maximum. Now consider

$$
\begin{aligned}
\mathrm{M}_p(\mathcal{S}; \boldsymbol{w}) &= \mathrm{M}_p(\mathcal{S}' + \boldsymbol{\varepsilon}; \boldsymbol{w}) && \text{\textsc{Definition of } } \boldsymbol{\varepsilon} \\
&\leq \mathrm{M}_p(\mathcal{S}' + \boldsymbol{\varepsilon}_+; \boldsymbol{w}) && \text{\textsc{Monotonicity}} \\
&\leq \mathrm{M}_p(\mathcal{S}'; \boldsymbol{w}) + \mathrm{M}_p(\boldsymbol{\varepsilon}_+; \boldsymbol{w}) && \text{\textsc{Item 2}} \\
&\leq \mathrm{M}_p(\mathcal{S}'; \boldsymbol{w}) + \mathrm{M}_p(\left| \mathcal{S} - \mathcal{S}' \right|; \boldsymbol{w}) \ , && \text{\textsc{Monotonicity}}
\end{aligned}
$$

where here Monotonicity refers to monotonicity of $\mathrm{M}_p(\mathcal{S}; \boldsymbol{w})$ in each $\mathcal{S}_i$, i.e., axiom 1. By symmetry, we have $\mathrm{M}_p(\mathcal{S}'; \boldsymbol{w}) \leq \mathrm{M}_p(\mathcal{S}; \boldsymbol{w}) + \mathrm{M}_p(\left| \mathcal{S} - \mathcal{S}' \right|; \boldsymbol{w})$, which implies the result.

We now show item 4. First note the special cases of $p \in \pm\infty$ follow by convexity of the maximum ($p = \infty$) and concavity of the minimum ($p = -\infty$).

Now, note that for $p \geq 1$, by concavity of $\sqrt[p]{\cdot}$, Jensen's inequality gives us

$$\mathrm{M}_1(\mathcal{S}; \boldsymbol{w}) = \sum_{i=1}^g \boldsymbol{w}_i \mathcal{S}_i = \underbrace{\sum_{i=1}^g \boldsymbol{w}_i \sqrt[p]{\mathcal{S}_i^p} \leq \sqrt[p]{\sum_{i=1}^g \boldsymbol{w}_i \mathcal{S}_i^p}}_{\text{\textsc{Definition of Convexity}}} = \mathrm{M}_p(\mathcal{S}; \boldsymbol{w}) \ ,$$

i.e., convexity, and similarly, for $p \leq 1$, $p \neq 0$, by convexity of $\sqrt[p]{\cdot}$, we have

$$\mathrm{M}_1(\mathcal{S}; \boldsymbol{w}) = \sum_{i=1}^g \boldsymbol{w}_i \mathcal{S}_i = \underbrace{\sum_{i=1}^g \boldsymbol{w}_i \sqrt[p]{\mathcal{S}_i^p} \geq \sqrt[p]{\sum_{i=1}^g \boldsymbol{w}_i \mathcal{S}_i^p}}_{\text{\textsc{Definition of Concavity}}} = \mathrm{M}_p(\mathcal{S}; \boldsymbol{w}) \ .$$

Similar reasoning, now by convexity of $\exp(\cdot)$, shows the case of $p = 0$. $\square$

We now show theorem 2.5.

**Theorem 2.5** (Aggregator Function Properties). Suppose aggregator function $\mathrm{M}(\mathcal{S}; \boldsymbol{w})$, and assume arbitrary sentiment vector $\mathcal{S} \in \mathbb{R}_{0+}^g$ and probability vector $\boldsymbol{w} \in \mathbb{R}_+^g$. If $\mathrm{M}(\cdot; \cdot)$ satisfies (subsets of) the aggregator-function axioms (see definition 2.2), then $\mathrm{M}(\cdot; \cdot)$ exhibits the following properties.

1. *Identity*: Axioms 6-7 imply that $\forall \alpha \in \mathbb{R}_{0+}: \mathrm{M}(\alpha \boldsymbol{1}; \boldsymbol{w}) = \alpha$.

2. *Debreu-Gorman Factorization*: Axioms 1-5 imply $\exists p \in \mathbb{R}$, strictly-monotonically-increasing continuous $F : \mathbb{R} \to \mathbb{R}_{0+}$ s.t.

$$\mathrm{M}(\mathcal{S}; \boldsymbol{w}) = F\left(\sum_{i=1}^{g} \boldsymbol{w}_i f_p(\mathcal{S}_i)\right) \quad, \quad \text{with} \begin{cases} p = 0 & f_0(x) \doteq \ln(x) \\ p \neq 0 & f_p(x) \doteq \mathrm{sgn}(p) x^p \end{cases} .$$

3. *Power-Mean Factorization*: Axioms 1-7 imply $F(x) = f_p^{-1}(x)$, thus $\mathrm{M}(\mathcal{S}; \boldsymbol{w}) = \mathrm{M}_p(\mathcal{S}; \boldsymbol{w})$.
4. *Fair Welfare*: Axioms 1-5 and 8 imply $p \in (-\infty, 1]$.
5. *Fair Malfare*: Axioms 1-5 and 9 imply $p \in [1, \infty)$.

*Proof.* First note that item 1 is an immediate consequence of axioms 6-7 (multiplicative linearity and unit scale).

We now note that item 2 is the celebrated Debreu-Gorman theorem, extended to weighted aggregator functions. The proof strategy for this extension is essentially to define two infinite sequences of rational weightings $\boldsymbol{w}^{j\downarrow}$ and $\boldsymbol{w}^{j\uparrow}$ that both converge to $\boldsymbol{w}$, each of which may be characterized by the *unweighted* Debreu-Gorman theorem, to then show that $\mathrm{M}(\mathcal{S}; \boldsymbol{w}^{j\downarrow}) \leq \mathrm{M}(\mathcal{S}; \boldsymbol{w}) \leq \mathrm{M}(\mathcal{S}; \boldsymbol{w}^{j\uparrow})$ for each sequence index $j$, and finally to apply standard continuity and limit properties to conclude the desideratum. In particular, the lower and upper weighting sequences $\boldsymbol{w}_i^{\downarrow}$ and $\boldsymbol{w}^{\uparrow}$ are composed entirely of weights that are *binary fractions*, the weighted symmetry axiom allows us to equate aggregate values with *rational weights* with aggregate values of a *larger unweighted* population, which is then characterized via the standard (unweighted) Debreu-Gorman theorem.

The (classical) Debreu-Gorman theorem describes the *unweighted case*, and is based on the *unweighted symmetry axiom*, whereas we seek to show the *weighted case*, i.e.,

$$\underbrace{\mathrm{M}(\mathcal{S}; \boldsymbol{w}) = F\left(\sum_{i=1}^{g} f_p(\mathcal{S}_i)\right)}_{\text{UNWEIGHTED}}, \underbrace{\mathrm{M}(\mathcal{S}; \boldsymbol{w}) = F\left(\sum_{i=1}^{g} \boldsymbol{w}_i f_p(\mathcal{S}_i)\right)}_{\text{WEIGHTED}}, \quad \text{with} \begin{cases} p = 0 & f_0(x) \doteq \ln(x) \\ p \neq 0 & f_p(x) \doteq \mathrm{sgn}(p) x^p \end{cases} .$$

Note that the unweighted case is equivalent to taking $\boldsymbol{w} = \langle \frac{1}{g}, \frac{1}{g}, \ldots, \frac{1}{g} \rangle$, but we omit these weight terms above, as they can be factored into the strictly-monotonically-increasing continuous function $F(\cdot)$. The unweighted case holds by the standard Debreu-Gorman theorem, as our weighted symmetry axiom generalizes the standard unweighted symmetry axiom, i.e., $\forall$ permutations $\pi$ over $\{1, \ldots, g\}$: $\mathrm{M}(\mathcal{S}) = \mathrm{M}(\pi(\mathcal{S}))$. To show the general case, a more sophisticated argument is necessary involving the *continuity* (axiom 3) of $\mathrm{M}(\mathcal{S}; \boldsymbol{w})$ w.r.t. $\boldsymbol{w}$ and the *weighted symmetry* axiom (2).

We now begin the reweighting argument proper. Suppose WLOG that $(\mathcal{S}; \boldsymbol{w})$ are jointly permuted s.t. $\mathcal{S}_1 \leq \mathcal{S}_2 \leq \cdots \leq \mathcal{S}_g$; this is always possible via the weighted symmetry axiom (2), e.g., for some permutation $\pi$, $\mathrm{M}(\langle 3, 2, 1 \rangle, \langle \frac{1}{2}, \frac{1}{6}, \frac{1}{3} \rangle) = \mathrm{M}(\pi(\langle 3, 2, 1 \rangle), \pi(\langle \frac{1}{2}, \frac{1}{6}, \frac{1}{3} \rangle)) = \mathrm{M}(\langle 1, 2, 3 \rangle, \langle \frac{1}{3}, \frac{1}{6}, \frac{1}{2} \rangle)$. Now, define lower and upper weighting sequences

$$\boldsymbol{w}_i^{j\downarrow} \doteq \min\left(1 - \sum_{i'=1}^{i-1} \boldsymbol{w}_{i'}^{j\downarrow}, \frac{\lceil \boldsymbol{w}_i 2^j \rceil}{2^j}\right) \quad, \quad \text{and } \boldsymbol{w}_i^{j\uparrow} \doteq \min\left(1 - \sum_{i'=i+1}^{g} \boldsymbol{w}_{i'}^{j\uparrow}, \frac{\lceil \boldsymbol{w}_i 2^j \rceil}{2^j}\right) ,$$

respectively, as illustrated in figure A4, and note that their limits obey

$$\lim_{j \to \infty} \boldsymbol{w}^{j\downarrow} = \boldsymbol{w} = \lim_{j \to \infty} \boldsymbol{w}^{j\uparrow} ,$$

and furthermore, by continuity of $\mathrm{M}(\mathcal{S}; \boldsymbol{w})$ in $\boldsymbol{w}$,

$$\lim_{j \to \infty} \mathrm{M}(\mathcal{S}; \boldsymbol{w}^{j\downarrow}) = \mathrm{M}(\mathcal{S}; \boldsymbol{w}) = \lim_{j \to \infty} \mathrm{M}(\mathcal{S}; \boldsymbol{w}^{j\uparrow}) .$$

Now, note that *cumulative weights* obey the simple relationship for all $i \in \{1, \ldots, g\}$:

$$\sum_{i'=1}^{i} \boldsymbol{w}_{i'}^{0\downarrow} \geq \sum_{i'=1}^{i} \boldsymbol{w}_{i'}^{1\downarrow} \geq \cdots \geq \sum_{i'=1}^{i} \boldsymbol{w}_{i'}^{\infty\downarrow} = \sum_{i'=1}^{i} \boldsymbol{w}_{i'} = \sum_{i'=1}^{i} \boldsymbol{w}_{i'}^{\infty\uparrow} \geq \cdots \geq \sum_{i'=1}^{i} \boldsymbol{w}_{i'}^{1\uparrow} \geq \sum_{i'=1}^{i} \boldsymbol{w}_{i'}^{0\uparrow} ,$$

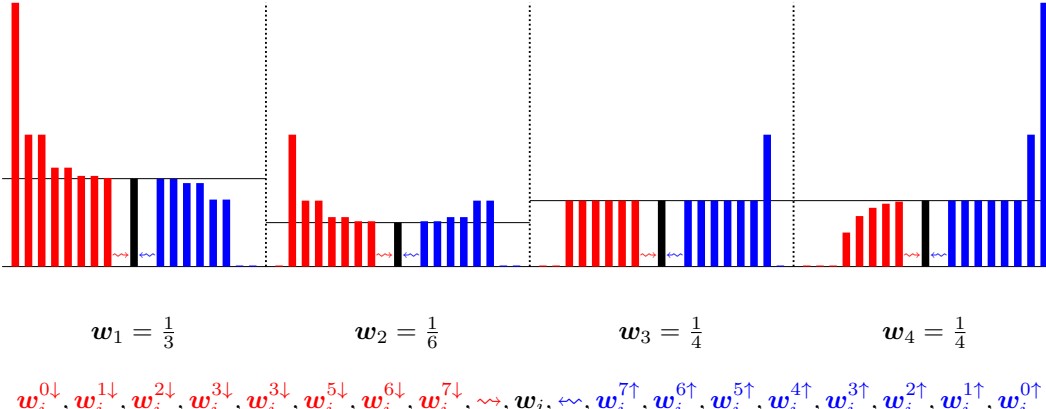

$$\boldsymbol{w}_1 = \tfrac{1}{3} \qquad \boldsymbol{w}_2 = \tfrac{1}{6} \qquad \boldsymbol{w}_3 = \tfrac{1}{4} \qquad \boldsymbol{w}_4 = \tfrac{1}{4}$$

$$\boldsymbol{w}_i^{0\downarrow}, \boldsymbol{w}_i^{1\downarrow}, \boldsymbol{w}_i^{2\downarrow}, \boldsymbol{w}_i^{3\downarrow}, \boldsymbol{w}_i^{3\downarrow}, \boldsymbol{w}_i^{5\downarrow}, \boldsymbol{w}_i^{6\downarrow}, \boldsymbol{w}_i^{7\downarrow}, \rightsquigarrow, \boldsymbol{w}_i, \leftsquigarrow, \boldsymbol{w}_i^{7\uparrow}, \boldsymbol{w}_i^{6\uparrow}, \boldsymbol{w}_i^{5\uparrow}, \boldsymbol{w}_i^{4\uparrow}, \boldsymbol{w}_i^{3\uparrow}, \boldsymbol{w}_i^{2\uparrow}, \boldsymbol{w}_i^{1\uparrow}, \boldsymbol{w}_i^{0\uparrow}$$

Figure A4: Illustration of lower and upper reweighting sequences $\boldsymbol{w}^{j\downarrow}$ and $\boldsymbol{w}^{j\uparrow}$, for weights vector $\boldsymbol{w} = \langle \tfrac{1}{2}, \tfrac{1}{6}, \tfrac{1}{4}, \tfrac{1}{4} \rangle$. Each $\boldsymbol{w}_i$ is shown in black, separated by vertical dotted lines, alongside the lower and upper reweighting sequences $\boldsymbol{w}_i^{j\downarrow}$ and $\boldsymbol{w}_i^{j\uparrow}$, for various values of $j$. Note that with increasing $j$, the binary fractional weights of the reweighting sequences more accurately approximate $\boldsymbol{w}$, and even by $j = 7$, the difference between $\boldsymbol{w}_i^{j\downarrow} \approx \boldsymbol{w} \approx \boldsymbol{w}^{j\uparrow}$ is essentially imperceptible above.

which is equivalent to *stochastic dominance* when each weights vector is viewed as a discrete random variable over $\{1, \ldots, g\}$. Intuitively, this is because the lower and upper weights sequences round weights progressively more accurately, in such a way that the lower sequence $\boldsymbol{w}_i^{j\downarrow}$ always transfers weight to higher group indices $i$ as sequence index $j$ increases, whereas the upper sequence $\boldsymbol{w}_i^{j\uparrow}$ always transfers weight to lower group indices $i$ as sequence index $j$ increases.

Now, by the weighted symmetry axiom (2), letting $\boldsymbol{w}^{j\uparrow\downarrow}$ refer generically to either $\boldsymbol{w}^{j\uparrow}$ or $\boldsymbol{w}^{j\downarrow}$,

$$\forall j : \ \mathrm{M}(\mathcal{S}; \boldsymbol{w}^{j\uparrow\downarrow}) = \mathrm{M}\left( \langle \underbrace{\mathcal{S}_1, \ldots, \mathcal{S}_1}_{2^j \boldsymbol{w}_1^{j\uparrow\downarrow} \text{ Copies}}, \underbrace{\mathcal{S}_2, \ldots, \mathcal{S}_2}_{2^j \boldsymbol{w}_2^{j\uparrow\downarrow} \text{ Copies}}, \ldots, \underbrace{\mathcal{S}_g, \ldots, \mathcal{S}_g}_{2^j \boldsymbol{w}_g^{j\uparrow\downarrow} \text{ Copies}} \rangle, \underbrace{i \mapsto \frac{1}{2^j}}_{\text{Uniform Weights}} \right) \ .$$

From here, by the monotonicity axiom (1), as well as the above argument regarding cumulative weights, and the fact that $\mathcal{S}$ was assumed to be ascending (and remains so above), we have

$$\mathrm{M}(\mathcal{S}; \boldsymbol{w}^{0\downarrow}) \leq \mathrm{M}(\mathcal{S}; \boldsymbol{w}^{1\downarrow}) \leq \cdots \leq \mathrm{M}(\mathcal{S}; \boldsymbol{w}^{\infty\downarrow}) = \mathrm{M}(\mathcal{S}; \boldsymbol{w}) = \mathrm{M}(\mathcal{S}; \boldsymbol{w}^{\infty\uparrow}) \leq \cdots \leq \mathrm{M}(\mathcal{S}; \boldsymbol{w}^{1\uparrow}) \leq \mathrm{M}(\mathcal{S}; \boldsymbol{w}^{0\uparrow}) \ .$$

In particular, monotonicity and weighted symmetry imply the inequalities relating the lower and upper weighting sequences, and continuity in $\boldsymbol{w}$ implies the equalities between the malfare values of (limits of) the weighting sequences and $\boldsymbol{w}$.

Now, by the (unweighted) Debreu-Gorman theorem, we have that there exists some $p \in \mathbb{R}$, and some strictly-monotonically-increasing continuous function $F(\cdot)$, s.t. for all $j$,

$$\mathrm{M}(\mathcal{S}^{j\uparrow\downarrow}; \boldsymbol{w}) = F\big( \mathrm{M}_p(\mathcal{S}^{j\uparrow\downarrow}; \boldsymbol{w}) \big) \ .$$

Substituting this result into the above yields the desideratum.

We now show item 3. This result is essentially a corollary of item 2, hence the dependence on axioms 1-5. In particular, we need only derive the canonical form of $F(\cdot)$ under this axiomatization, which is quite straightforward when one considers the case of $\mathcal{S} = \mathbf{1}$. By item 1 (as we now assume axioms 6-7), and axiom 6 itself, for all $p \neq 0$, we have

$$\alpha = \alpha \mathrm{M}(\mathbf{1}; \boldsymbol{w}) = \mathrm{M}(\alpha \mathbf{1}; \boldsymbol{w}) = F\left( \sum_{i=1}^{g} \boldsymbol{w}_i f_p(\alpha) \right) = F\big( \mathrm{sgn}(p) \alpha^p \big) \ .$$

From here, we have $\alpha = F\big( \mathrm{sgn}(p) \alpha^p \big)$, thus $F^{-1}(u) = \mathrm{sgn}(p) u^p$, and consequently, $F(v) = \sqrt[p]{\mathrm{sgn}(p) v}$.

Taking $p = 0$, and applying similar logic to the above, gets us

$$\alpha = \alpha \mathrm{M}(\mathbf{1}; \boldsymbol{w}) = \alpha \mathrm{M}(\alpha \mathbf{1}; \boldsymbol{w}) = F\left(\sum_{i=1}^{g} \boldsymbol{w}_i \ln(\alpha)\right) = F(\ln \alpha) \ ,$$

from which it is clear that $F^{-1}(u) = \ln(u) \implies F(v) = \exp(v)$.

For all values of $p \in \mathbb{R}$, substituting the values of $f_p$ and $F(\cdot)$ into item 2 yields $\mathrm{M}(\mathcal{S}; \boldsymbol{w}) = \mathrm{M}_p(\mathcal{S}; \boldsymbol{w})$ by definition.

We now show items 4 and 5. These properties follow directly from item item 3 and item 4, alongside the observation that the Pigou-Dalton transfer principle (axiom 8) implies $\mathrm{M}(\mathcal{S}; \boldsymbol{w})$ is *concave* in $\mathcal{S}$, whereas the negated Pigou-Dalton transfer principle (axiom 9) implies $\mathrm{M}(\mathcal{S}; \boldsymbol{w})$ is *convex* in $\mathcal{S}$.   $\square$

## B.2  FPAC-Learning

We now work towards showing theorem 3.2. We first show some lemmas in service of the main result.

**Lemma B.1** (Statistical Estimation). Suppose probability distributions $\mathcal{D}_{1:g}$, samples $\boldsymbol{z}_{1:g,1:m} \sim \mathcal{D}_1^m \times \cdots \times \mathcal{D}_g^m$, aggregator function $\mathrm{M}(\cdot; \cdot)$ obeying *monotonicity*, probability vector $\boldsymbol{w} \in \mathbb{R}_+^g$, and some sentiment-function $f$. Now take *sentiment vector* $\mathcal{S} \in \mathbb{R}_{0+}^g$ such that $\mathcal{S}_i = \mathbb{E}_{\mathcal{D}_i}[f]$, and *empirical sentiment value estimate* $\hat{\mathcal{S}} \in \mathbb{R}_{0+}^g$ such that $\hat{\mathcal{S}}_i \doteq \hat{\mathbb{E}}_{\boldsymbol{z}_i}[f]$. If it holds with probability at least $1 - \delta$ that $\forall i \in 1, \ldots, g : \hat{\mathcal{S}}_i - \boldsymbol{\varepsilon}_i \le \mathcal{S}_i \le \hat{\mathcal{S}}_i + \boldsymbol{\varepsilon}_i$, then with said probability, we have

$$\mathrm{M}_p(\mathbf{0} \vee (\hat{\mathcal{S}} - \boldsymbol{\varepsilon}); \boldsymbol{w}) \le \mathrm{M}_p(\mathcal{S}; \boldsymbol{w}) \le \mathrm{M}_p(\hat{\mathcal{S}} + \boldsymbol{\varepsilon}; \boldsymbol{w}) \ ,$$

where $\boldsymbol{a} \vee \boldsymbol{b}$ denotes the (elementwise) maximum.

*Proof.* This result follows from the assumption, and the *monotonicity* axiom (i.e., adding/subtracting $\varepsilon$ can not decrease/increase the aggregator function value, respectively). The minimum with $0$ on the LHS is *valid* simply because, by definition, sentiment values are nonnegative, and is *necessary*, as $\mathrm{M}_p(\cdot; \boldsymbol{w})$ is in general undefined with negative inputs.   $\square$

We now state a standard result[4] in uniform convergence theory (see, e.g., [18, 26]).

**Lemma B.2** (Emprical Rademacher Bounds). Suppose $\ell : (\mathcal{Y} \times \mathcal{Y}) \to [0, r]$, and $\mathcal{H} \subseteq \mathcal{X} \to \boldsymbol{y}$. Suppose also $\boldsymbol{z} \sim \mathcal{D}^m$. Then, with probability at least $1 - \delta$ over choice of $\boldsymbol{z}$, we have

$$\sup_{h \in \mathcal{H}} \left| \hat{\mathrm{R}}(h; \ell, \boldsymbol{z}) - \mathrm{R}(h; \ell, \mathcal{D}) \right| \le 2\hat{\mathfrak{R}}_m(\ell \circ \mathcal{H}, \boldsymbol{z}) + 3r\sqrt{\frac{\ln \frac{1}{\delta}}{2m}} \ .$$

With these results, we are now ready to show theorem 3.2.

**Theorem 3.2** (Generalization Guarantees for Malfare Estimation). Suppose hypothesis class $\mathcal{H} \subseteq \mathcal{X} \to \mathcal{Y}$, bounded *loss function* $\ell : (\mathcal{Y} \times \mathcal{Y}) \to [0, r]$, and per-group samples $\boldsymbol{z}_i \sim \mathcal{D}_i^m$. Then, with probability at least $1 - \delta$ over choice of $\boldsymbol{z}$, it holds *simultaneously* for all *fair malfare functions* $\mathrm{\Lambda}(\cdot; \cdot)$ (i.e., $\mathrm{\Lambda}_p(\cdot; \cdot)$ for $p \ge 1$) and *probability vectors* $\boldsymbol{w} \in \mathbb{R}_+^g$, that

$$\sup_{h \in \mathcal{H}} \left| \mathrm{\Lambda}\big(i \mapsto \mathrm{R}(h; \ell, \mathcal{D}_i); \boldsymbol{w}\big) - \mathrm{\Lambda}\big(i \mapsto \hat{\mathrm{R}}(h; \ell, \boldsymbol{z}_i); \boldsymbol{w}\big) \right| \le \mathrm{\Lambda}\left(i \mapsto 2\hat{\mathfrak{R}}_m(\ell \circ \mathcal{H}, \boldsymbol{z}_i) + 3r\sqrt{\frac{\ln \frac{g}{\delta}}{2m}}; \boldsymbol{w}\right) \ .$$

*Proof.* This result follows from the above statistical lemmas, and the contraction property of malfare. In particular, note that by lemma B.2, with probability at least $1 - \delta$, by union bound it holds that

$$\forall i \in 1, \ldots, g : \forall h \in \mathcal{H} : \left| \hat{\mathrm{R}}(h; \ell, \boldsymbol{z}_i) - \mathrm{R}(h; \ell, \mathcal{D}) \right| \le 2\hat{\mathfrak{R}}_m(\ell \circ \mathcal{H}, \boldsymbol{z}_i) + 3r\sqrt{\frac{\ln \frac{g}{\delta}}{2m}} \ .$$

The desideratum then follows via lemma B.1 and the *contraction property* of malfare, i.e., theorem 2.4 item 3. Note that the above statement on risk holds *in probability*, but the rest of the argument holds *deterministically*, hence the *with probability at least* $1 - \delta$ qualifier holds over all fair malfare functions and weighting probability vectors *simultaneously*.   $\square$

---

[4]Note cautiously that we bound the *absolute supremum deviation*, which incurs no union bound penalty, as we define the *empirical Rademacher average* itself with the absolute value built in.

We now show theorem 3.7.

**Theorem 3.7** (Realizable Reductions). Suppose loss function $\ell$ and hypothesis class $\mathcal{H}$. Then

1. $(\mathcal{H}, \ell) \in \mathrm{PAC}^0 \implies (\mathcal{H}, \ell) \in \mathrm{FPAC}^0$; and     2. $(\mathcal{H}, \ell) \in \mathrm{PAC}^0_{\mathrm{Poly}} \implies (\mathcal{H}, \ell) \in \mathrm{FPAC}^0_{\mathrm{Poly}}$.

We construct a(n) (efficient) FPAC-learner for $(\mathcal{H}, \ell)$ by noting that there exists some $\mathcal{A}'$ with sample-complexity $\mathrm{m}_{\mathcal{A}'}(\varepsilon, \delta, d)$ and time complexity $\mathrm{t}_{\mathcal{A}'}(\varepsilon, \delta, d)$ to PAC-learn $(\mathcal{H}, \ell)$, and taking $\mathcal{A}(\mathcal{D}_{1:g}, \boldsymbol{w}, \mathbb{M}, \varepsilon, \delta, d) \doteq \mathcal{A}'(\mathrm{mix}(\mathcal{D}_{1:g}), \frac{\varepsilon}{g}, \delta, d)$, where $\mathrm{mix}(\mathcal{D}_{1:g})$ denotes the uniformly-weighted mixture of distributions $\mathcal{D}_{1:g}$. Then $\mathcal{A}$ FPAC-learns $(\mathcal{H}, \ell)$, with sample-complexity $\mathrm{m}_{\mathcal{A}}(\varepsilon, \delta, d, g) = \mathrm{m}_{\mathcal{A}'}(\frac{\varepsilon}{g}, \delta, d)$, and time-complexity $\mathrm{t}_{\mathcal{A}}(\varepsilon, \delta, d, g) = \mathrm{t}_{\mathcal{A}'}(\frac{\varepsilon}{g}, \delta, d)$.

*Proof.* We first show the *correctness* of the constructed FPAC-learner $\mathcal{A}$. Suppose $\hat{h} \leftarrow \mathcal{A}(\mathcal{D}_{1:g}, \boldsymbol{w}, \mathbb{M}, \varepsilon, \delta, d)$. Then, with probability at least $1 - \delta$, by the PAC-learnability guarantee of $\mathcal{A}'$, we have

$$\mathbb{M}_p(i \mapsto \mathrm{R}(h; \ell, \mathcal{D}_i)); \boldsymbol{w}) \leq \mathbb{M}_\infty\big(i \mapsto \mathrm{R}(h; \ell, \mathcal{D}_i); i \mapsto \tfrac{1}{g}\big)$$
$$\leq g\mathbb{M}_1\big(i \mapsto \mathrm{R}(h; \ell, \mathcal{D}_i); i \mapsto \tfrac{1}{g}\big)$$
$$= g\mathrm{R}\big(h; \ell, \mathrm{mix}(\mathcal{D}_{1:g})\big) \leq g\tfrac{\varepsilon}{g} = \varepsilon .$$

We thus may conclude that $(\mathcal{H}, \ell)$ is efficiently *realizable-PAC-learnable* by $\mathcal{A}'$, with *sample complexity* $\mathrm{m}_{\mathcal{A}}(\varepsilon, \delta, d, g) = \mathrm{m}_{\mathcal{A}'}(\frac{\varepsilon}{g}, \delta, d)$. Similarly, if $\mathcal{A}$ has polynomial runtime, then so too does $\mathcal{A}'$, thus we may also conclude efficiency. $\square$

We now show theorem 3.10.

**Theorem 3.10** (Fundamental Theorem of Fair Statistical Learning). Suppose $\ell$ such that $\forall \mathcal{H}$ : $(\mathcal{H}, \ell) \in \mathrm{PAC}^0 \implies (\mathcal{H}, \ell) \in \mathrm{UC}$. Then $\forall \mathcal{H}$, the following are equivalent:

1. $\forall d \in \mathbb{N}$: $(\ell, \mathcal{H}_d)$ has the generalized uniform convergence property.
2. Any EMM rule is an agnostic-FPAC learner for $(\ell, \mathcal{H})$, thus $(\ell, \mathcal{H})$ is agnostic-FPAC learnable.
3. Any EMM rule is a realizable-FPAC learner for $(\ell, \mathcal{H})$, thus $(\ell, \mathcal{H})$ is realizable-FPAC learnable.

*Proof.* First note that $1 \implies 2$ is a rather straightforward consequence of the definition of uniform convergence and the contraction property of fair malfare functions (theorem 2.4 item 3). In particular, take $m \doteq \mathrm{m}_{\mathrm{UC}}(\ell \circ \mathcal{H}_d, \frac{\varepsilon}{2}, \frac{\delta}{g})$. By union bound, this implies that with probability at least $1 - \delta$, taking samples $\boldsymbol{z}_{1:g,1:m} \sim \mathcal{D}_1^m \times \cdots \times \mathcal{D}_g^m$, we have

$$\forall i \in \{1, \ldots, g\} : \sup_{h \in \mathcal{H}_d} \Big| \mathrm{R}(h; \ell, \mathcal{D}_i) - \hat{\mathrm{R}}(h; \ell, \boldsymbol{z}_i) \Big| \leq \frac{\varepsilon}{2} .$$

Consequently, as $\mathbb{M}(\cdot; \boldsymbol{w})$ is $1\text{-}\|\cdot\|_\infty\text{-}|\cdot|$-Lipschitz in risk (see lemma B.1), it holds with probability at least $1 - \delta$ that

$$\forall h \in \mathcal{H}_d : \Big| \mathbb{M}\big(i \mapsto \hat{\mathrm{R}}(h; \ell, \boldsymbol{z}_i); \boldsymbol{w}\big) - \mathbb{M}\big(i \mapsto \mathrm{R}(h; \ell, \mathcal{D}_i); \boldsymbol{w}\big) \Big| \leq \frac{\varepsilon}{2} .$$

Now, for EMM-optimal $\hat{h}$, and malfare-optimal $h^*$, we apply this result twice to get

$$\mathbb{M}\big(i \mapsto \mathrm{R}(\hat{h}; \ell, \mathcal{D}_i); \boldsymbol{w}\big) \leq \mathbb{M}\big(i \mapsto \hat{\mathrm{R}}(\hat{h}; \ell, \boldsymbol{z}_i); \boldsymbol{w}\big) + \tfrac{\varepsilon}{2}$$
$$\leq \mathbb{M}\big(i \mapsto \hat{\mathrm{R}}(h^*; \ell, \boldsymbol{z}_i); \boldsymbol{w}\big) + \tfrac{\varepsilon}{2}$$
$$\leq \mathbb{M}\big(i \mapsto \mathrm{R}(h^*; \ell, \mathcal{D}_i); \boldsymbol{w}\big) + \varepsilon .$$

Therefore, under uniform convergence, the EMM algorithm agnostic FPAC learns $(\mathcal{H}, \ell)$ with finite sample complexity $\mathrm{m}_{\mathcal{A}}(\varepsilon, \delta, d, g) = g \cdot \mathrm{m}_{\mathrm{UC}}(\ell \circ \mathcal{H}_d, \frac{\varepsilon}{2}, \frac{\delta}{g})$, completing $1 \implies 2$.

Now, $2 \implies 3$ holds, as realizable learning is a special case of agnostic learning.

It remains only to show that $3 \implies 1$, i.e., if $\mathcal{H}$ is realizable FPAC learnable, then $\mathcal{H}$ has the uniform convergence property. In general, the question is rather subtle, but here the assumption "suppose $\ell$ such that $(\mathcal{H}, \ell) \in \mathrm{PAC}^0 \implies (\mathcal{H}, \ell) \in \mathrm{UC}$" does most of the work. In particular, as PAC-learning is a special case of FPAC-learning, we have

$$(\mathcal{H}, \ell) \in \mathrm{FPAC} \implies (\mathcal{H}, \ell) \in \mathrm{PAC} ,$$

then applying the assumption yields $(\mathcal{H}, \ell) \in \mathrm{UC}$. $\square$

## B.3  Efficient FPAC-Learning

We now show theorem 3.11.

**Theorem 3.11** (Efficient FPAC Learning via Convex Optimization). Suppose each hypothesis space $\mathcal{H}_d \in \mathcal{H}$ is indexed by $\Theta_d \subseteq \mathbb{R}^{\mathrm{Poly}(d)}$, i.e., $\mathcal{H}_d = \{h(\cdot;\theta) \mid \theta \in \Theta_d\}$, s.t. (Euclidean) $\mathrm{Diam}(\Theta_d) \in \mathrm{Poly}(d)$, and $\forall x \in \mathcal{X}, \theta \in \Theta_d$, $h(x;\theta)$ can be evaluated in $\mathrm{Poly}(d)$ time, and $\tilde{\theta} \in \mathbb{R}^{\mathrm{Poly}(d)}$ can be Euclidean-projected onto $\Theta_d$ in $\mathrm{Poly}(d)$ time. Suppose also $\ell$ such that $\forall x \in \mathcal{X}, y \in \mathcal{Y} : \theta \mapsto \ell(y, h(x;\theta))$ is a *convex function*, and suppose Lipschitz constants $\lambda_\ell, \lambda_{\mathcal{H}} \in \mathrm{Poly}(d)$ and some norm $\|\cdot\|_{\mathcal{Y}}$ over $\mathcal{Y}$ s.t. $\ell$ is $\lambda_\ell \|\cdot\|_{\mathcal{Y}} \text{-}|\cdot|$-Lipschitz in $\hat{y}$, i.e.,

$$\forall y, \hat{y}, \hat{y}' \in \mathcal{Y} : \left|\ell(y,\hat{y}) - \ell(y,\hat{y}')\right| \leq \lambda_\ell \|\hat{y} - \hat{y}'\|_{\mathcal{Y}} \quad,$$

and also that each $\mathcal{H}_d$ is $\lambda_{\mathcal{H}} \|\cdot\|_2 \text{-}\|\cdot\|_{\mathcal{Y}}$-Lipschitz in $\theta$, i.e.,

$$\forall x \in \mathcal{X}, \theta, \theta' \in \Theta_d : \left\|h(x;\theta) - h(x;\theta')\right\|_{\mathcal{Y}} \leq \lambda_{\mathcal{H}} \|\theta - \theta'\|_2 \quad.$$

Finally, assume $\ell \circ \mathcal{H}_d$ exhibits $\varepsilon$-$\delta$ *uniform convergence* with sample complexity $\mathrm{m}_{\mathrm{UC}}(\varepsilon, \delta, d) \in \mathrm{Poly}(\frac{1}{\varepsilon}, \frac{1}{\delta}, d)$. It then holds that, for arbitrary initial guess $\theta_0 \in \Theta_d$, for any group distributions $\mathcal{D}_{1:g}$, group weights $\boldsymbol{w}$, and fair malfare function $\mathcal{M}(\cdot;\cdot)$, the algorithm (see algorithm 1)

$$\mathcal{A}(\mathcal{D}_{1:g}, \boldsymbol{w}, \mathcal{M}(\cdot;\cdot), \varepsilon, \delta, d) \doteq \mathcal{A}_{\mathrm{PSG}}\big(\ell, \mathcal{H}_d, \theta_0, \mathrm{m}_{\mathrm{UC}}(\cdot,\cdot,d), \mathcal{D}_{1:g}, \boldsymbol{w}, \mathcal{M}(\cdot;\cdot), \varepsilon, \delta\big)$$

FPAC-learns $(\mathcal{H}, \ell)$ with sample complexity $\mathrm{m}(\varepsilon, \delta, d, g) = g \cdot \mathrm{m}_{\mathrm{UC}}(\frac{\varepsilon}{3}, \frac{\delta}{g}, d)$, and (training) time-complexity $\in \mathrm{Poly}(\frac{1}{\varepsilon}, \frac{1}{\delta}, d, g)$, thus $(\mathcal{H}, \ell) \in \mathrm{FPAC}_{\mathrm{Poly}}$.

*Proof.* We now show that this projected-subgradient method construction of $\mathcal{A}$ requires $\mathrm{Poly}(\frac{1}{\varepsilon}, \frac{1}{\delta}, d, g)$ time to identify an $\varepsilon$-$\delta$-$\mathcal{M}_p(\cdot;\cdot)$-optimal $\tilde{\theta} \in \Theta_d$, and thus FPAC-learns $(\mathcal{H}, \ell)$. This essentially boils down to showing that (1) the empirical malfare objective is *convex* and *Lipschitz continuous*, and (2) that algorithm 1 runs sufficiently many projected-subgradient update steps, with appropriate step size, on a sufficiently large training set, to yield the appropriate guarantees, and that each step of the projected-subgradient method, of which there are polynomially many, itself requires polynomial time.

First, note that by theorem 2.5 items 3 and 5, we may assume that $\mathcal{M}(\cdot;\cdot)$ can be expressed as a $p$-power-mean with $p \geq 1$; thus henceforth we refer to it as $\mathcal{M}_p(\cdot;\cdot)$. Now, recall that the empirical malfare objective (given $\theta \in \Theta_d$ and training sets $\boldsymbol{z}_{1:g}$) is defined as

$$\mathcal{M}_p\big(i \mapsto \hat{\mathrm{R}}(h(\cdot;\theta); \ell, \boldsymbol{z}_i); \boldsymbol{w}\big) \quad.$$

We first show that empirical malfare is convex in $\theta \in \Theta_d$. By assumption and positive linear closure, $\hat{\mathrm{R}}(h(\cdot;\theta'); \ell, \boldsymbol{z}_i)$ is convex in $\theta \in \Theta_d$. The objective of interest is the composition of $\mathcal{M}_p(\cdot;\boldsymbol{w})$ with this empirical risk evaluated on each of the $g$ training sets $\boldsymbol{z}_{1:g}$. By theorem 2.4 item 4, $\mathcal{M}_p(\cdot;\boldsymbol{w})$ is convex $\forall p \in [1,\infty]$ in $\mathbb{R}_{0+}^g$, and by the monotonicity axiom, it is monotonically increasing. Composition of a monotonically increasing convex function on $\mathbb{R}_{0+}^g$ with convex functions on $\Theta_d$ yields a convex function, thus we conclude the empirical malfare objective is convex in $\Theta_d$.

We now show that empirical malfare is Lipschitz-continuous. Now, note that for any $p \geq 1$, $\boldsymbol{w}$,

$$\forall \mathcal{S}, \mathcal{S}' : \left|\mathcal{M}_p(\mathcal{S};\boldsymbol{w}) - \mathcal{M}_p(\mathcal{S}';\boldsymbol{w})\right| \leq 1\|\mathcal{S} - \mathcal{S}'\|_\infty \quad,$$

i.e., $\mathcal{M}_p(\cdot;\boldsymbol{w})$ is $1\|\cdot\|_\infty\text{-}|\cdot|$-Lipschitz in *empirical risks* (see theorem 2.4 item 3), and thus by Lipschitz composition, we have Lipschitz property

$$\forall \theta, \theta' \in \Theta_d : \left|\mathcal{M}_p\big(i \mapsto \hat{\mathrm{R}}(h(\cdot;\theta); \ell, \boldsymbol{z}_i); \boldsymbol{w}\big) - \mathcal{M}_p\big(i \mapsto \hat{\mathrm{R}}(h(\cdot;\theta'); \ell, \boldsymbol{z}_i); \boldsymbol{w}\big)\right| \leq \lambda_\ell \lambda_{\mathcal{H}} \|\theta - \theta'\|_2 \quad.$$

We now show that algorithm 1 FPAC-learns $(\mathcal{H}, \ell)$. As above, take $m \doteq \mathrm{m}_{\mathrm{UC}}(\frac{\varepsilon}{3}, \frac{\delta}{g}, d)$. Our algorithm shall operate on a training sample $\boldsymbol{z}_{1:g,1:m} \sim \mathcal{D}_1^m \times \cdots \times \mathcal{D}_g^m$.

First note that evaluating a subgradient (via forward finite-difference estimation or automated sub-differentiation) requires $(\mathrm{Dim}(\Theta_d) + 1)m$ evaluations of $h(\cdot;\cdot)$, which by assumption is possible in $\mathrm{Poly}(d, m) = \mathrm{Poly}(\frac{1}{\varepsilon}, \frac{1}{\delta}, d, g)$ time.

The projected subgradient method produces $\tilde{\theta}$ approximating the empirically-optimal $\hat{\theta}$ such that [see 27]

$$f(\tilde{\theta}) \leq f(\hat{\theta}) + \frac{\|\theta_0 - \hat{\theta}\|_2^2 + \Lambda^2 \alpha^2 n}{2\alpha n} \leq \frac{\mathrm{Diam}^2(\Theta_d) + \Lambda^2 \alpha^2 n}{2\alpha n} \quad,$$

for $\Lambda$-$\|\cdot\|_2$-$|\cdot|$-Lipschitz objective $f$, thus taking $\alpha \doteq \frac{\mathrm{Diam}(\Theta_d)}{\Lambda\sqrt{n}}$ yields

$$f(\tilde{\theta}) - f(\hat{\theta}) \leq \frac{\mathrm{Diam}(\Theta_d)\Lambda}{\sqrt{n}} \quad.$$

As shown above, $\Lambda = \lambda_\ell \lambda_\mathcal{H}$, thus we may guarantee *optimization error*

$$\varepsilon_{\mathrm{opt}} \doteq f(\hat{\theta}) - f(\theta^*) \leq \frac{\varepsilon}{3}$$

if we take iteration count

$$n \geq \frac{9\,\mathrm{Diam}^2(\Theta_d)\lambda_\ell^2\lambda_\mathcal{H}^2}{\varepsilon^2} = \left(\frac{3\,\mathrm{Diam}(\Theta_d)\lambda_\ell\lambda_\mathcal{H}}{\varepsilon}\right)^2 \in \mathrm{Poly}(\tfrac{1}{\varepsilon}, d) \quad.$$

As each iteration requires $m \cdot \mathrm{Poly}(d) \subseteq \mathrm{Poly}(\frac{1}{\varepsilon}, \frac{1}{\delta}, d, g)$ time, the projected-subgradient method identifies an $\frac{\varepsilon}{3}$-empirical-malfare-optimal $\tilde{\theta} \in \Theta_d$ in $\mathrm{Poly}(\frac{1}{\varepsilon}, \frac{1}{\delta}, d, g)$ time.

As $m$ was selected to ensure $\frac{\varepsilon}{3}$-$\frac{\delta}{g}$ uniform convergence, we thus have that by uniform convergence, and union bound (over $g$ groups), with probability at least $1 - \delta$ over choice of $\boldsymbol{z}_{1:g}$, we have

$$\forall i \in \{1, \ldots, g\}, \theta \in \Theta_d : \left|\mathbb{M}_p(i \mapsto \hat{\mathrm{R}}(h(\cdot; \theta); \ell, \boldsymbol{z}_i); \boldsymbol{w}) - \mathbb{M}_p(i \mapsto \mathrm{R}(h(\cdot; \theta); \ell, \mathcal{D}_i); \boldsymbol{w})\right| \leq \frac{\varepsilon}{3} \quad.$$

Combining *estimation* and *optimization* errors, we get that with probability at least $1 - \delta$, the approximate-EMM-optimal $h(\cdot; \tilde{\theta})$ obeys

$$\begin{aligned}
\mathbb{M}_p(i \mapsto \mathrm{R}(h(\cdot; \tilde{\theta}); \ell, \mathcal{D}_i); \boldsymbol{w}) &\leq \mathbb{M}_p(i \mapsto \hat{\mathrm{R}}(h(\cdot; \tilde{\theta}); \ell, \boldsymbol{z}_i); \boldsymbol{w}) + \tfrac{\varepsilon}{3} \\
&\leq \mathbb{M}_p(i \mapsto \hat{\mathrm{R}}(h(\cdot; \hat{\theta}); \ell, \boldsymbol{z}_i); \boldsymbol{w}) + \tfrac{2\varepsilon}{3} \\
&\leq \mathbb{M}_p(i \mapsto \hat{\mathrm{R}}(h(\cdot; \theta^*); \ell, \boldsymbol{z}_i); \boldsymbol{w}) + \tfrac{2\varepsilon}{3} \\
&\leq \mathbb{M}_p(i \mapsto \mathrm{R}(h(\cdot; \theta^*); \ell, \mathcal{D}_i); \boldsymbol{w}) + \varepsilon \quad.
\end{aligned}$$

We may thus conclude that $\mathcal{A}$ FPAC learns $\mathcal{H}$ with sample complexity $gm = g \cdot \mathrm{m}_{\mathrm{UC}}(\frac{\varepsilon}{3}, \frac{\delta}{g}, d)$. Furthermore, as the entire operation requires polynomial time, we have $(\mathcal{H}, \ell) \in \mathrm{PAC}_{\mathrm{Poly}}$. $\square$