# OpenReview forum: "An Axiomatic Theory of Provably-Fair Welfare-Centric Machine Learning"
_NeurIPS.cc/2021/Conference — NeurIPS 2021 Poster_

### Official Review · Reviewer_VZaR · 2021-07-08

**Rating:** 7
**Confidence:** 2

**Summary:**

The authors consider malfaire metrics,
measuring overall societal harm, with axiomatic justification via
standard axioms of welfare, and cast fair ML as malfare minimization
over the risk values (expected losses) of each group. this is not
equivalent to simply defining utility as negative loss and maximizing welfare.


**Limitations And Societal Impact:**

Yes.

**Main Review:**

The paper tells us that malfare metrics have different properties to welfare
metrics and there may be applications where minimizing malfare is possible
but maximising welfare is not. I am not sure whether this is common
or whether it just occurs for pathological situations. Nevertheless,
it does seem important to know this. It would be useful (if possible) to give a
brief mathematical explanation for the difference between welfare and malfare.
Is the only important difference possible Lipschitz-discontinuity or are there
others?

Results concerning PAC learning are generalised to the case of
F(for fair)PAC learning (which takes into account malfare).
This is clearly solid work backed up by proofs in the supplementary material.

I just have one serious niggling doubt: axiom 6 is presented without any
obvious justification. All other axioms have appropriate citations showing that
they are standard (or at least have been previously proposed).
Axiom 6 is essential to show that the malfare function must be a power means function.
(One could also probably find applications where some of the other axioms
maybe do not apply, but they do seem to be less contentious
than axiom 6). Axiom 6 is a strengthening of a traditional cardinal welfare axiom,
but is this strengthening really justified?

Figure 1: Is there a simple summary of what this figure is telling us?
I think that results improve (in terms of fairness) as p increases, but the
important question is why? Is there a one-line explanation for this?

line 320: This is a "generalisation" in the sense that the definition is slacker,
which is not necessarily a good thing: the mean may converge when the frequency does not.
Is this a possibly important difference or not? In Theorem 3.9 you mean generalised UC
not UC, so don't put parentheses around "generalised".

Minor points:
- In certain figs (in the supplementary material), eg race-01,
the y-axis has 1 several times which is clearly a typo.
- line 105: write pre-existing instead of preëxisting
- line 163: multiplicative linearity is axiom 6 in your list, not 5.
- line 164: typo?: "nats or of cross-entropy loss"
- line 167: "milquetoast" seems to be an inappropriate adjective. Perhaps 'weak' would be better.
- Figure 1 (legend): what happens if b_i is close to 0?
- Def. 3.3 use words instead of \forall, \exists
- line 279: acronym Agn not defined.
- line 291: careful with line numbers in middle of lines
- Def. 3.7 perhaps give a reference for Uniform Convergence.
- Fig. 2: what do dashed implications mean?

**Time Spent Reviewing:**

5

---

> ### Author Response · Authors · 2021-08-10
> **Author Response**
>
> We think the reviewer for their helpful feedback and insightful questions. We respond below.
>
> *The paper tells us that malfare metrics have different properties to welfare metrics and there may be applications where minimizing malfare is possible but maximising welfare is not. I am not sure whether this is common or whether it just occurs for pathological situations. Nevertheless, it does seem important to know this. It would be useful (if possible) to give a brief mathematical explanation for the difference between welfare and malfare. Is the only important difference possible Lipschitz-discontinuity or are there others?*
>
> For the purposes of statistical estimation, Lipschitz-continuity is probably the most important difference.
> However, despite their common functional form, there are some substantial differences between the $p \leq 1$ and $p \geq 1$ cases, except ironically at the endpoints of $p = 1$ and $p \in \pm \infty$, where welfare  and malfare or utility and disutility really do just flips signs.
> Qualitatively, it is somewhat difficult to describe these differences, though we argue that the PAC learning framework is a nice illustration of what can be done with malfare, and that the corresponding learnability concept for welfare cannot exist, due to non-Lipschitz and general inestimability properties of some welfare functions.
> We also note that both welfare and malfare require nonnegative (semireal) sentiment values, and their behavior is most distorted as these values approach 0 or extend across a broad range.
> The distortion around 0 can not be symmetrical for the two cases, because for utility, this is distortion of the *least favorable* sentiment, whereas for utility, it is distortion of the *most favorable* sentiment.
>
>
> *I just have one serious niggling doubt: axiom 6 is presented without any obvious justification. All other axioms have appropriate citations showing that they are standard (or at least have been previously proposed). Axiom 6 is essential to show that the malfare function must be a power means function. (One could also probably find applications where some of the other axioms maybe do not apply, but they do seem to be less contentious than axiom 6). Axiom 6 is a strengthening of a traditional cardinal welfare axiom, but is this strengthening really justified?*
>
> This is an excellent point, and there is some subjectivity to the "naturalness" of axioms. The most direct answer we can give is that multiplicative linearity is essentially the same as assuming the dimensions welfare match the dimensions of the utility, which is a reasonable axiom, since it allows comparison between sentiment values and their malfare or welfare summaries, as well as between functions with various $p$. This is not wholly convincing, but when coupled with the fact that, up to isomorphism, no functions are lost due to this axiom, we consider it a reasonable standardization.
>
> An alternative perspective is that it is quite common in cardinal welfare theory to assume additive separability, and we consider this far less justified than multiplicative linearity.  The only advantage we see to additive separability is marginally simpler computation, and furthermore even after assuming additive separability, additional axioms are needed to get the canonical form specifying the additive and multiplicative constants.  Thus we argue that while the matter is somewhat subjective, we think that at least a better case can be made for multiplicative linearity than additive separability, and the latter is already widely accepted.
>
> *Figure 1: Is there a simple summary of what this figure is telling us? I think that results improve (in terms of fairness) as p increases, but the important question is why? Is there a one-line explanation for this?*
>
> The details are subtle, but the simplest explanation is that as $p$ increases, the relative impact of the high-risk groups, which tend to be small and traditionally marginalized groups, on the loss function increases.
> Thus the optimal classifier must make more trade-offs in favor of these groups, which increases fairness. In the limit as $p \to \infty$, we reach egalitarianism, where we select the minimax-optimal classifier, which can be thought of as robustly learning over multiple groups, which provides further intuition as to why the situation becomes more fair relative to the $p=1$ (utilitarian) case, which is just a straight (weighted) average.
>
> *line 320: This is a "generalisation" in the sense that the definition is slacker, which is not necessarily a good thing: the mean may converge when the frequency does not. Is this a possibly important difference or not? In Theorem 3.9 you mean generalised UC not UC, so don't put parentheses around "generalised".*
>
> We're not sure we follow: the standard uniform convergence definition applies only to {$0,1$} valued functions, for which expectations and frequencies are the same. Our generalization applies to arbitrary real-valued functions, and thus coincides for indicator function families.  It merely expands the scope to real-valued families, which allows us to analyze learning with loss functions other than unweighted 0-1 loss.
>
>
>
> *Figure 1 (legend): what happens if b_i is close to 0?*
>
> As $b_i$ approaches $0$, the relative importance of a group (as measured by their weight) approaches $\infty$, which is of course a problem in general, but this is just an experimental setup here to correct for the fact that some groups are far easier to predict than others, and the actual values of $b_i$ range from $11.7\%$ to $26.9$ (percent), thus the actual impact is only a relative weighting of up to a factor $2.3$. This weighting of course linearly increases Rademacher averages, standard deviations, and ranges, and thus linearly impacts the tail bounds of theorem 3.2.
> We considered optimizing ``excess risk’’ rather than weighted risk, where we considered the difference between empirical risk on the shared model and on a  "reference model" trained just for the group in question (which would never be greater than the "shared risk").  This idea would avoid the issue of division by zero, but ultimately we decided to go with the simpler reweighting approach.

---

### Official Review · Reviewer_vUPX · 2021-07-14

**Rating:** 7
**Confidence:** 4

**Summary:**

This paper investigates an important challenge in welfare-centric fair ML, that is, how to connect the standard loss minimization view of ML with the utility maximization view of welfare-based fairness. The authors define malfare to measure societal harm: malfare is a complementary metric of welfare, and it is justified with axioms of cardinal welfare (and extensions to malfare). Utilizing the malfare metric, the authors cast fair ML that seeks equitable risk distribution to groups as malfare minimization over the expected losses of all groups. They further build up on the malfare minimization framework to extend PAC learning to fair-PAC (FPAC) learning, which is proposed as a formal notion of fair learnability. Their analysis of FPAC learning shows that standard PAC-learners may be converted to FPAC-learners under some conditions, and these connections further yield theoretical results on the statistical and computational learnability with FPAC learners.

**Limitations And Societal Impact:**

The authors acknowledged the limitations of their work in several aspects, such as the condition needed to prove Theorem 3.9, the difficulty in extending the analysis to fair welfare maximization. I think the limitations are adequately discussed, and as these limitations are non-trivial to address, it is reasonable to leave as future work as indicated by the authors.

As the paper studies fair ML, it naturally seeks to address potential negative societal impacts of ML. One aspect that may be useful to discuss is the welfare implication of EMM in practice. The authors state that EMM should be used when there is no explicit way to convert loss to utility, nevertheless, one may still wonder how the learner trained with EMM performs with respect to different utility definitions. For instance, if the predictions about whether income is above or below $50k per year are used to guide certain decisions, equitable risk distribution in the predictor does not necessarily guarantee desirable utility distribution in these decisions.

**Main Review:**

This paper studies a new problem in fair machine learning. The adopted methodology is a novel combination of the cardinal welfare theory from economics, welfare-centric methods in fair ML and the well-established PAC learning theory. In my opinion, the results are significant as they fill important theoretical gaps in fair ML. The paper focuses on welfare-centric fair ML, which is a relatively new thread of fair ML that uses a social welfare function to represent fairness. Compared to the dominant parity based fairness notions used in ML, the welfare-centric view could provide a broader perspective on fairness driven by the well-beings of stakeholders and represent a wide range of principled fairness concepts. In existing literature, these potential advantages tend to be undermined by the lack of theoretical justification from the learning aspect, that is, there are no general learning performance guarantees on welfare-centric fair ML. This paper contributes FPAC learning as one possible framework to offer solid theoretical justifications, thus, in my opinion, greatly expands the potentials of welfare-metric fair ML.

The paper is well organized and clearly written. The authors provide sufficient introduction of the background and motivations of their task by citing related works. They also gave clear explanations of what distinguish the work from previous research. The presented technical results are solid, and the proofs and derivations are correct as far as I could verify.

I find the paper well written for the most part, and I only have a few minor questions and suggestions.
1. From my understanding, the power mean formulation puts welfare and malfare on a spectrum from equality to inequality controlled by the power $p$. Do the authors think it is reasonable to interpret malfare as inequality measure? If so, I think it might be interesting to compare the FPAC framework or Empirical malfare minimization (EMM) with the results in [24]. On a related note, a potential reference to include is ‘An Axiomatic Theory of Fairness in Network Resource Allocation’ by Lan et al.: this paper adopts a similar axiomatic view on fairness and derive a mean-type format for fairness measures; if I recall correctly, the axioms they consider coincide with Axiom 1-5 used in the submission.
2. In the appendix of [11], the authors discussed a mapping between loss minimization to social welfare maximization. Although their welfare function definition is different from the one used in the submission, I think it is worth mentioning this result in the ‘Related work’.
3. The authors highlighted that EMM is more fitting in ML contexts since there often is no clearly neutral way to define utility based on loss. While I agree with this statement, I think in many cases the decision makers designing the learning task would have the expertise or authority to know what utility functions to consider. Certainly the utility definitions could be context dependent, but one may argue that it is desirably so to capture context-specific welfare considerations. Can the authors include brief examples of when utility definitions would be hard to define?
4. Can the authors provide some intuition on which of the welfare/malfare axioms and welfare/malfare function properties appear to be indispensable for developing the theoretically sound FPAC learning framework? Along similar line, how hard will it be to extend the analysis to other welfare measures?

Minor comments:
1. Figure 1: the test hinge risk is shown as higher than the training hinge risk reversed pattern for the ‘Black’ group, while the other groups all have the reverse pattern. Is this figure correct?
2. Theorem 2.5 Condition 2: is the summation term $\sum_{i=1}^g f_p(\mathcal{S}_i)$ missing the weight vector?
3. Appendix proof of Theorem 2.4: line 558 and 559, convexity and concavity appear to be mixed up.

**Time Spent Reviewing:**

8

---

> ### Author Response · Authors · 2021-08-10
> **Author Response**
>
> We thank the reviewer for their insightful questions and valuable connections to the literature. We will do our best to include discussion of these references in the camera-ready copy, and are excited to pursue these connections further in future work.  We respond to specific points below.
>
> (1)
> In some sense, power means can be thought of as inequality indices, as the three values of the arithmetic mean, the $p$-power mean, and the $p$-generalized entropy index have two degrees of freedom. Thus, in some sense considering the power mean is similar to methods that constrain an entropy index and maximize utilitarian welfare; we have some results on this, but didn't think they'd we're as interesting as what we chose to include in the limited space available.
>
> (2)
>
> I believe we covered this paper on lines 74 and 75 in the context of direct welfare maximization?  Or perhaps you are referring to their SVM formulation?  We found this a bit tricky to compare to, as the actual optimization was on a relaxation of the loss function of interest, so it was difficult to directly interpret the connection between loss and welfare.  It is also true that they give a correspondence (Proposition 1) between welfare maximization and loss minimization, but the loss functions are not specified, so it's not clear how to use this result to optimize welfare.
>
> Furthermore, while reweighting loss functions seems like a tempting optimization strategy, we aren't sure it's useful in general.  Even for very simple problems, like choosing whether to allocate error as (2, 0), (0, 2), or (1, 1) between two groups, only for weighting (.5, .5) is the fair allocation optimal, and even then, it is not uniquely optimal, as all allocations have weighted utility 1, so it’s not clear how reweighted utilitarian welfare can be used to optimize other power means.
>
> (3)
>
> Utility functions are often hard to define, because they quantify things that are hard to measure.
> For instance money, income, and goods can easily be quantified with utility functions and loss thereof with disutility functions, but something like happiness or subjective appreciation or satisfaction with an event are much more difficult to quantify. The nice thing about loss functions (e.g., the cross entropy loss) is that many come from principled decision-theoretic analysis, and don't require an analysis of how the system in question affects its users.
> We note that understanding the downstream effects and implications of a model are extremely important for fairness, and our model is compatible with such an approach, but also that we identify a fundamental problem even with very simplistic cases where accuracy or a differential loss values affect each group in the same way.
>
> Furthermore, more fundamentally, utility and disutility values are both assumed to be semireal, so utility functions can only be applied when there is a lower bound on how good something can be, whereas disutility instead requires a lower bound on how bad something can be, so in this sense, there is a topological difference.
>
>
> (4)
>
> We could very well dispense with the welfare and malfare axioms, and begin the analysis with a definition of FPAC learning that mandates we minimize with uniform sample complexity over some class of malfare functions, or maximize with uniform sample complexity over some class of welfare functions. The work would be sound, but less interesting, because the foundation would be unstable, in the sense that one could immediately ask *why* a particular class of functions was chosen, so the whole concept feels artificial, and less fundamental.
> We believe we make a strong case for the class we selected, and furthermore we show that it does lead to interesting (nondegenerate and nontrivial) learnability results, whereas if we had instead sought to maximize welfare or minimize or maximize welfare or malfare under additive separability, no uniform sample complexity results would be possible. Moreover, analyzing such a class and coming up with interesting reductions seems intractable without more restrictions, and the general framework is rather like the Seldonian learner of [25], so we think the argument for a specific class of interest is quite important.
>
>
>
> This being said, restricting to other welfare concepts, such as the generalized Gini social welfare may be interesting, and in that particular case each member of the welfare family is Lipschitz-continuous and everything can be estimated efficiently, as in the malfare case.
> In general, all we require for statistical estimation is Lipschitz continuity (even that can be slightly relaxed in some cases). Our convex optimization analysis also requires that malfare functions are convex and welfare functions are concave, although these are also implied by the Pigou-Dalton and Anti Pigou-Dalton transfer principle axioms.
>
>
>
>
>
> *Minor comments:*
>
> *(1)    Figure 1: the test hinge risk is shown as higher than the training hinge risk reversed pattern for the ‘Black’ group, while the other groups all have the reverse pattern. Is this figure correct?*
>
>
> We also observed this phenomenon and determined that it was not due to a software or plotting error.
> The phenomenon is not stable under multiple runs of the algorithm, but rather is a consequence of the fact that the malfare is most strongly impacted by the larger white group for small $p$, and the higher risk native American group for large $p$, so at no point is the loss incurred on the Black group a strong enough factor on the selection of $\hat{h}$ that it overfits appreciably.
> With 5 groups, it is likely that one or more would sometimes sample an unlucky training set that is more challenging than the test set, particularly with our small sample sizes.  Coupled with a lack of overfitting on this group, we see training performance worse than test performance.
>
> We considered applying known bounce on Rademacher averages for SVMs along with theorem 3.2, to upper and lower bound the training losses for each group, as well as the malfare, however for the sample sizes in question the bounds were vacuous, so that this occurs is not surprising.

---

> > ### Comment · Reviewer_vUPX · 2021-08-26
> > **Thank you for the detailed response**
> >
> > Thank you for the detailed responses and for answering all my questions. As a quick follow up to (2), I meant to ask for further elaboration on the high-level comparison stated in lines 74-75. To this end, I think the argument given in (2) is helpful and it makes the distinctions between [11] and the submission more clear.

---

### Official Review · Reviewer_Wv61 · 2021-07-16

**Rating:** 7
**Confidence:** 3

**Summary:**

The paper is concerned with welfare-theoretic fair machine learning. Instead of enforcing statistical parity as a constraint, the paper instead casts fair learning as malfare minimization. Based on the axioms of cardinal welfarism, axioms for malfare are defined. Theoretical guarantees based on learning theory are established for malfare minimization.

**Limitations And Societal Impact:**

I think the impacts were adequately discussed.

**Main Review:**

-I think the paper addresses an important topic and the way it casts fair machine learning is interesting and important.

-I did not find the paper easy to read and it felt that too many results were packed in a limited number of pages.

-It seems more natural to show that the axioms imply Th 2.5 then introduce definition 2.3 for the power mean.

-Line 129: doesn't axiom 6 imply 5.

-Axiom 4 (Def 2.2): the symbols should have some explanation.

-Def 2.3: I think there is a typo for M_{\infty} , it should be S_i not S_g also brackets for the set.

-The paper mentions that we cannot simply define utility as negative loss, but I did not find a strong argument in the paper against that. Why can't a naive transformation that shifts the utility to positive values work?

-The  sentiment vector at the end takes only a fixed form which is the risk for each population. I assume that it not  possible to use learning theory guarantees without the sentiment taking on this form. But isn't there now a loss of generality to the formulation, couldn't different groups have different ways for evaluating sentiment just the way two individuals assign a different utility to the same item?

-I think the representation would improve if at least in the main body, the brief (informal) statements of theorem are given instead.

-I think the simplex notation is usually used for probability vectors, so w \in \Delta^{g-1}





**Time Spent Reviewing:**

3

---

> ### Author Response · Authors · 2021-08-10
> **Author Response**
>
> We thank the reviewer for their detailed and careful feedback. You are correct, Axiom 6 implies Axiom 5, and $S_g$ is a typo.  The organizational feedback is also appreciated. We address specific points below.
>
>
> *The paper mentions that we cannot simply define utility as negative loss, but I did not find a strong argument in the paper against that. Why can't a naive transformation that shifts the utility to positive values work?*
>
> We will add more detail on the salient differences between malfare and welfare in the final paper, but for now we give some intuition below.
> In some cases, one can in principle do this, although some desiderata are in general lost.
>
> We want utility and risk to be means over groups, so the transformation function has to be linear (furthermore, if we are allowed nonlinear transformations, then any power mean can be converted into any other power mean, as $M_{q}({\bf \ell}, {\bf w}) = \sqrt[q/p]{M_{p}(i \mapsto {\bf \ell}_{i}^{q/p}; {\bf w})}$, so at this point we lose the meaning and fairness properties of the power mean).
>
> For bounded loss functions with domain $[0, r]$, we can define the transformation as $x \mapsto \beta - x$ for any $\beta \geq r$.
> However, even for such a simple transformation, we actually lose some of the cardinal welfare axioms, in this case *independence of common scale*.
> Furthermore, as $\beta$ increases, we lose fairness in the sense that
>
> $
> \displaystyle\forall p \in \mathbb{R}: \lim_{\beta \to \infty} M_{p}(\beta \pm {\bf \ell}, {\bf w}) - \beta = \pm M_{1}({\bf \ell}; {\bf w}) \enspace,
> $
>
> thus everything becomes utilitarian, so such transformations can severely damage the fairness properties of the selected malfare / welfare function.
>
>
> *The sentiment vector at the end takes only a fixed form which is the risk for each population. I assume that it not possible to use learning theory guarantees without the sentiment taking on this form. But isn't there now a loss of generality to the formulation, couldn't different groups have different ways for evaluating sentiment just the way two individuals assign a different utility to the same item?*
>
> If we understand your question correctly you are asking whether each group could have their sentiment, or equivalently, their own loss function? This is actually possible within our framework, and our experiments do this, albeit for the simplistic case of weighted loss.
>
> In particular, all of our setups can handle mixed loss functions by attaching an extra piece of information to each sample $x_i$, which is used by $\ell$ to determine which loss function to use.
> This immediately yields the appropriate results for the Rademacher bounds, and can also be applied to the FPAC setup.  In the FPAC setting, the construction is actually slightly more general: technically each sample of each group can have its own loss function specified by this auxiliary information, which may actually be desirable, but it does also make sense to restrict each group to share the same fixed loss function.
> We considered making this explicit in the definition of FPAC-learning (i.e., allowing per-group loss functions), but we concluded that it was difficult enough to follow all of the parameters as it is.

---

### Official Review · Reviewer_aRtP · 2021-07-23

**Rating:** 6
**Confidence:** 4

**Summary:**

The paper follows the axioms of cardinal welfare, in conjunction with an introduced notion of malfare which quantifies the amount of societal harm rather than welfare which quantifies overall benefit. This allows the authors to cast fair machine learning problems as risk minimization problems, where they aim to minimize the malfare function. The function, which follows the set of axioms, is a power mean function, defined over the set of groups of interest, weighing the loss on each, to output a combined, single, value. Casting the function as malfare allows them in turn to use a specific characteristic of the proposed formulation – contraction – which is crucial in establishing their generalization result, through bounding the values of the base loss function, and the use of Rademacher complexity. Finally, they show that for their notion of fair learning through the malfare function, in the agnostic case and under standard convex optimization assumptions, it is possible to efficiently learn through the subgradient method. The authors also perform an experimental evaluation of their suggested approach on the adult dataset.

**Limitations And Societal Impact:**

I believe that the authors have addressed the limitations of their work properly. I would appreciate further discussion regarding their suggested objective of fairness and in which cases the authors think it should be used.

**Main Review:**

I think that the paper does a nice job of formalizing and reasoning for the fairness objective they advocate for, and further – by showing that their formulation allows for generalization through the contraction property, which I thought was a nice observation. In terms of practicality, I think the proposed approach makes a good contribution for cases where we believe fairness should be measured in terms of accuracy in the different groups. In such cases, the algorithm they propose should be considered as one alternative in the fairness toolkit. In terms of overall novelty, my assessment is that the paper makes a moderate contribution. I think that the main theoretical contribution is through the observation that certain loss aggregation functions may allow for generalization using the contraction attribute, which leads to uniform sample complexity bounds.

A main weakness that I see in the approach is that it weighs loss, in its general form, over the different groups. We have already seen that in many of the settings where fairness is a concern, false positives and false negatives can have entirely different effects on the individual (beneficial or detrimental), thus bundling them together and minimizing the overall loss may possibly result in an accumulation of mistakes of different types on different groups, which by itself may be problematic.

The presentation of the paper can be slightly improved, especially section 2, where the flow I thought was a bit confusing.
The submission appears to be technically sound, although I have only glanced at proofs in the appendix. The paper cites most of the relevant prior work, but could add a short discussion regarding existing group-based fairness techniques.

Questions to authors:
1. Can the suggested approach be extended to also take into account the types of losses (false positives, false negatives)? How would your approach handle cases where the two types of losses have very different implications?
2. Assuming access to an oracle for standard 0-1 ERM in the agnostic case, would it be possible to harness this “building block” to optimize for your proposed malfare objective?

**Time Spent Reviewing:**

10

---

> ### Author Response · Authors · 2021-08-10
> **Author Response**
>
> We thank the reviewer for their valuable feedback and insightful questions.
>
>
> (1) WV61 asks a similar question about per-group loss functions.
>
> We considered this situation when developing FPAC learning and the rest of our methodology, however it is not clearly described in the paper.  Our experiments, however, do use a specific type of weighted loss, and our Rademacher bounds, FPAC learning algorithms, and other results do apply to arbitrary (even per-group weighted) loss functions, however the construction is a bit too subtle to be left to subtext.  We will be sure to explicitly construct such an example in the paper to clarify that our methodology is sufficient to handle weighted loss functions.
>
> In particular, all of our setups can handle weighted losses by attaching an extra piece of information to each $x_i$, which is used by $\ell$ to weight appropriately.
> This immediately yields the appropriate results for the Rademacher bounds, and can also be applied to the FPAC setup.  In the FPAC setting, the construction is actually slightly more general: technically *each sample*, rather than just each group, can have its own loss function (i.e., reweighting) specified by this auxiliary information, which may actually be desirable, but it does also make sense to restrict each group to share the same fixed loss function.
>
> We considered making this explicit in the definition of FPAC-learning (i.e., allowing per-group loss functions), but we concluded that it was difficult enough to follow all of the parameters as it is.  Furthermore, as far as we can tell, bounded discrete reweightings in classification preserve FPAC and efficient FPAC learnability (at constant factor costs), and our convexity arguments is also valid, so long as each individual loss function is convex, so we did not think explicitly generalizing the learning model was actually necessary.
>
>
> (2) ERM Oracle:
>
> This is a difficult question to answer directly, and there is quite some subtlety to it.
> On one level, this oracle solves an NP-hard problem, and thus vastly increases our computational capacity, and we believe it would be possible to optimize welfare given access to a nondeterministic Turing machine.
>
> However, if the dimension of the input problem for the oracle is restricted, this computational advantage is much smaller, as the oracle can be implemented in roughly $O(dm^d)$ work.  In this case, we conjecture the answer is no, at least not in the sense of “yes, with the following satisfying reduction,” mostly because in our research, we did try to construct a generic FPAC learner in terms of reweightings of PAC learners / ERM oracles, and met with little success.
> We think it might be possible with continuous loss functions under appropriate constraints, but in the discrete case it seems hopeless as there exists a natural one-dimensional 2-group binary classification problem for which the egalitarian-optimal (or any $p > 1$ optimal) classifier doesn't uniquely optimize any weighted combination of the two groups.
>
> For example, under perfect disagreement between the two groups (i.e., when one is red, the other is blue, and vice-versa), then there are three cases: the reweighting is even, in which case all classifiers are equally good (in expectation), or one or the other group is more highly weighted, in which case the weighted-optimal classifier is the same as the optimal classifier for just the higher-weight group.  The egalitarian-optimal is only optimal for the reweighting where all classifiers are equally accurate (inaccurate), so it hardly seems that the reweighting can help, even in this trivial example.

---

> > ### Comment · Reviewer_aRtP · 2021-08-26
> > **Thank you for the response**
> >
> > (1) I believe a short discussion about this should appear in the main text of the paper.
> > (2) In the second question, when asking about ERM oracle, I am referring to the commonly used abstraction of “oracle-efficiency”, which in practice is translated to assuming access to some successful heuristic for efficiently optimizing the 0-1 loss over a finite set of samples from the domain of the original problem. We would then of course be interested in the case where the size of input to this oracle is restricted in terms of the original problem. In any case, the answer is clear.

---

### Decision · Program_Chairs · 2021-09-27

**Decision:**

Accept (Poster)

**Comment:**

The reviewers’ overall assessment of the paper was that (a) it proposes an interesting, relatively well-justified formulation of fairness through the concept of malware minimization. (2) The ensuing learnability analysis is non-trivial and the proposed algorithms can be practically informative.  Reviewers made several suggestions to the authors to improve the flow and exposition (in particular, it is important that the authors expand their discussion of the proposed axioms, in particular, axiom 6, in the main body of the paper). Assuming that the authors will reflect those changes in their next revision of the paper, I recommend acceptance.